# Tissue-resident macrophages actively suppress IL-1beta release via a reactive prostanoid/IL-10 pathway

Natacha Ipseiz[1,†], Robert J Pickering[1,†] iD, Marcela Rosas[1], Victoria J Tyrrell[1], Luke C Davies[1],
Selinda J Orr[1,2], Magdalena A Czubala[1], Dina Fathalla[1,3], Avril AB Robertson[4], Clare E Bryant[5,6] iD,
Valerie O'Donnell[1] & Philip R Taylor[1,3,*] iD

## Abstract

The alarm cytokine interleukin-1β (IL-1β) is a potent activator of the inflammatory cascade following pathogen recognition. IL-1β production typically requires two signals: first, priming by recognition of pathogen-associated molecular patterns leads to the production of immature pro-IL-1β; subsequently, inflammasome activation by a secondary signal allows cleavage and maturation of IL-1β from its pro-form. However, despite the important role of IL-1β in controlling local and systemic inflammation, its overall regulation is still not fully understood. Here we demonstrate that peritoneal tissue-resident macrophages use an active inhibitory pathway, to suppress IL-1β processing, which can otherwise occur in the absence of a second signal. Programming by the transcription factor Gata6 controls the expression of prostacyclin synthase, which is required for prostacyclin production after lipopolysaccharide stimulation and optimal induction of IL-10. In the absence of secondary signal, IL-10 potently inhibits IL-1β processing, providing a previously unrecognized control of IL-1β in tissue-resident macrophages.

**Keywords** IL-10; IL-1beta; macrophages; prostacyclin
**Subject Category** Immunology
The EMBO Journal (2020) 39: e103454

## Introduction

Interleukin-1β (IL-1β) is a pro-inflammatory cytokine, an alarmin which, once released into the extracellular milieu, triggers the inflammatory response. It is commonly accepted that a two-step mechanism is required for IL-1β production in mouse macrophages (MΦ). First, pathogen-associated molecular pattern (PAMP) recognition induces transcription and translation of the inactive pro-form of IL-1β (pro-IL-1β). A secondary signal, such as reactive oxygen species (ROS) (Nakahira et al, 2011; Zhou et al, 2011), crystals (Hornung et al, 2008) or potassium efflux (Petrilli et al, 2007), is then needed to induce the classical inflammasome assembly, composed of NOD-like receptor family, pyrin domain containing 3 (Nlrp3) and apoptosis-associated speck-like protein containing a CARD (ASC), also called PYCARD. Once assembled, the NLRP3 inflammasome activates caspase1 which in turn cleaves pro-IL-1β into its mature IL-1β form (Bryant & Fitzgerald, 2009; Dowling & O'Neill, 2012; Latz et al, 2013; Lamkanfi & Dixit, 2014). Despite intensive research, the mechanisms regulating IL-1β maturation and release (Lopez-Castejon & Brough, 2011; Martin-Sanchez et al, 2016) are still under discussion (Cullen et al, 2015; Evavold et al, 2018; Monteleone et al, 2018). Dysregulated IL-1β production has been associated with the development of many inflammatory and autoinflammatory diseases (Lamkanfi & Dixit, 2012, 2014; Yao et al, 2016; Mayer-Barber & Yan, 2017) such as cryopyrin-associated periodic syndromes (CAPS), type 2 diabetes (Jourdan et al, 2013), increased susceptibility to Crohn's disease (Villani et al, 2009) and intestinal inflammation (Shouval et al, 2016), gout (Joosten et al, 2010) and rheumatoid arthritis (Pascual et al, 2005).

Macrophages are part of the immune system's first line of defence. Initially simply categorized as phagocytes, evidence of their complexity has accumulated over the years (Ley et al, 2016). Resident peritoneal macrophages (pMΦ), a well-studied tissue macrophage population, have essential functions, including modulation of the inflammatory response after pathogen recognition (Dioszeghy et al, 2008; Spight et al, 2008; Leendertse et al, 2009) or injury

1 Systems Immunity Research Institute, Heath Park, Cardiff University, Cardiff, UK
2 Wellcome-Wolfson Institute for Experimental Medicine, School of Medicine, Dentistry and Biomedical Science, Queen's University Belfast, Belfast, UK
3 UK Dementia Research Institute at Cardiff, Cardiff University, Cardiff, UK
4 School of Chemistry and Molecular Biosciences, The University of Queensland, Brisbane, Qld, Australia
5 Immunology Catalyst Programme, GSK, Cambridge, UK
6 Department of Veterinary Medicine, University of Cambridge, Cambridge, UK
*Corresponding author. Tel: +44 02920687328; E-mail: taylorpr@cardiff.ac.uk
†These authors contributed equally to this work as first authors

(Uderhardt *et al*, 2019), phagocytosis of pathogens (Ghosn *et al*, 2010) and dying cells (Fond & Ravichandran, 2016), liver repair (Wang & Kubes, 2016; Rehermann, 2017) and maintenance of self-tolerance (Russell & Steinberg, 1983; Mukundan *et al*, 2009; Munoz *et al*, 2010; Uderhardt *et al*, 2012; Ipseiz *et al*, 2014; Majai *et al*, 2014; Carlucci *et al*, 2016). pMΦ are part of the first wave of response during peritonitis (Khameneh *et al*, 2017) and help ensure the survival of the host and the optimal clearance of the infection. Their efficiency is coupled to their optimal cytokine and chemokine secretion which have to be finely tuned, including IL-1β (Topley *et al*, 1996; Hautem *et al*, 2017). After the first inflammatory burst following PAMP recognition, macrophages dampen their inflammatory processes by producing anti-inflammatory molecules, such as IL-10 (Bogdan *et al*, 1991; Berlato *et al*, 2002; Saraiva & O'Garra, 2010). IL-10 protects against acute inflammation (Howard *et al*, 1993), and its loss has dramatic effects as observed in IL-10 deficient mice, which develop chronic enterocolitis (Kuhn *et al*, 1993; Krause *et al*, 2015). However, the regulatory control of IL-10 production by pMΦ (Liao *et al*, 2016) as well as its mode of action remain unclear. Additionally, after microbial stimulation, peritoneal macrophages rapidly release prostanoids, such as prostaglandin I2 (PGI2), also called prostacyclin (Brock *et al*, 1999). PGI2 is known to be generated in peritoneal macrophages following inflammatory stimulation, although it is poorly studied and neglected in this context (Yokode *et al*, 1988; Stewart *et al*, 1990; Wightman & Dallob, 1990). Despite the paradoxical role of PGI2 in inflammatory diseases (Stitham *et al*, 2011), synthetic analogues can decrease tumour necrosis factor (TNF) and induce IL-10 in human peripheral mononuclear cells *in vitro* (Eisenhut *et al*, 1993; Luttmann *et al*, 1999) and inhibit function of murine dendritic cells (Zhou *et al*, 2007), suggesting an active control of inflammation.

Here we have studied the inflammatory response of resident pMΦ that lack their specialized tissue-programming as a consequence of deletion of the tissue-specific transcription factor Gata6 (Rosas *et al*, 2014). While wild-type (WT) pMΦ need a secondary signal after lipopolysaccharide (LPS) stimulation to produce mature IL-1β, we show that the Gata6-deficient pMΦ do not, and they exhibit aberrant production of IL-1β after LPS stimulation. Using Gata6-KO^[mye] pMΦ, we identified a Gata6-PGI2-IL-10 axis as a major regulator of IL-1β processing in resident pMΦ. This axis actively inhibits IL-1β processing during response to a microbial stimulus in the absence of a second signal and thus ensures proportionate and finely regulated production of IL-1β in response to LPS.

# Results

## Gata6-deficient peritoneal macrophages exhibit dysregulated IL-1β release

We and others previously identified the transcription factor Gata6 as a major key regulator of tissue-resident peritoneal macrophage (pMΦ) specialization (Gautier *et al*, 2014; Okabe & Medzhitov, 2014; Rosas *et al*, 2014). To determine its role in the inflammatory function of pMΦ, we analysed the response of Gata6-WT and Gata6-KO^[mye] pMΦ after toll-like receptor (TLR) ligand stimulation. Surprisingly, we observed that ultra-pure LPS, a specific TLR4 agonist, induced the production of IL-1β by Gata6-KO^[mye] pMΦ in

the absence of an exogenous secondary signal (Fig 1A). LPS also induced significantly higher production of TNF by Gata6-KO^[mye] pMΦ as observed by ELISA (Fig 1B) and flow cytometry (Figs EV1 and 2A). Additionally, the effect of LPS on IL-1β and TNF production was found to be concentration (Fig 1C and D) and time-dependent (Fig 1E and F). Interestingly, inhibiting TNF with etanercept (a fusion protein composed of TNFR2 connected to a human IgG1 Fc tail) slightly reduced IL-1β production from Gata6-KO^[mye] pMΦ and blocking the IL-1 signalling pathway using an IL-1 receptor antagonist (rIL-1ra) when stimulating the cells with purified LPS did not dramatically change IL-1β secretion by Gata6-WT or Gata6-KO^[mye] pMΦ (Fig 1G). Stimulating the cells with recombinant TNF (recTNF) did not induce IL-1β production from either Gata6-WT or Gata6-KO^[mye] pMΦ (Fig 1H). These data suggest that neither IL-1 receptor nor TNF signalling greatly affected IL-1β production.

## Activation of the NLRP3 inflammasome by classical stimuli is independent of Gata6 programming

To investigate the mechanism behind the aberrant release of IL-1β by Gata6-KO^[mye] pMΦ following LPS stimulation, we analysed the components of the classical NLRP3 inflammasome pathway. Gata6-WT and KO^[mye] pMΦ showed similar levels of toll-like receptors (*Tlr*) expression, with an exception for *Tlr13* which appeared to be reduced in Gata6-KO^[mye] cells. *Tlr4* and *Cd14*, the receptor and co-receptor, respectively, for LPS were similarly expressed by Gata6-WT and KO^[mye] cells (Fig 2A). Cell surface expression analysis by flow cytometry however showed an increase in TLR2 and TLR4 in KO^[mye] cells (Figs 2B and EV2B). Gata6-WT and KO^[mye] pMΦ exhibited similar *Il1b* mRNA expression (Fig 2C) and comparable production of pro-IL-1β (Figs 2D and EV3A), 3 and 6 h after LPS stimulation, respectively. Interestingly, Gata6-KO^[mye] cells showed a significantly upregulated *Tnf* expression (Fig 2C), indicating a direct regulation of TNF production on a mRNA level rather than on a protein level. The pro-IL-1β expression was confirmed by flow cytometry analysis (Fig 2E). These results suggest that, despite an increased TLR4 expression in Gata6-KO^[mye] pMΦ, both Gata6-WT and KO^[mye] pMΦ have a similar response capacity to the primary signal LPS regarding the initiation of IL-1β production and that the aberrant release observed in the Gata6-KO^[mye] cells was likely due to a downstream dysregulation in pro-IL-1β processing. Further investigation revealed that Gata6-KO^[mye] pMΦ did not have increased *Nlrp3* mRNA expression compared to WT pMΦ (Fig 2F), as well as similar protein levels (Figs 2G and EV3B). To determine whether the classical Nlrp3 inflammasome was responsible for the IL-1β release in Gata6-KO^[mye] pMΦ, we stimulated Gata6-WT and KO^[mye] pMΦ with LPS in the presence of the specific Nlrp3 inhibitor MCC950 (Coll *et al*, 2015, 2019). We observed an abrogation of IL-1β secretion from Gata6-KO^[mye] cells (Fig 2H and I), confirming the essential role of Nlrp3 in the production of IL-1β by Gata6-KO pMΦ. In addition, Gata6-KO^[mye] cells showed upregulated mRNA expression of caspase1 (*Casp1*) (Fig 2J) but protein expression showed no significant difference (Figs 2K and EV3C). The selective caspase1 inhibitor Ac-YVAD-cmk blocked IL-1β secretion by LPS-stimulated Gata6-KO^[mye] pMΦ (Fig 2L). Interestingly, when first primed with LPS for 3 h and then stimulated with a secondary signal (ATP or nigericin) for 30 min, both Gata6-WT and KO^[mye]

pMΦ released comparable levels of IL-1β (Fig 2M). Confocal immunofluorescence analysis showed that both cell types were able to form ASC specks, a hallmark of the classical NLRP3 inflammasome assembly, when stimulated with LPS and ATP (Fig 2N). Overall, these data indicate a similar NLRP3 inflammasome capacity of both Gata6-WT and KO^mye pMΦ, suggesting that the aberrant IL-1β release observed in Gata6-KO^mye cells after LPS stimulation might be due to alteration of the regulatory mechanisms and not to the inflammasome machinery itself.

## Peritoneal-resident macrophages actively suppress IL-1β production

Based on our findings above, we next hypothesized that the Gata6-KO^mye pMΦ had a defect in an inhibitory pathway, present in the Gata6-WT cells, that basally constrains further processing of pro-IL-1β after LPS stimulation. To investigate this hypothesis, we co-cultured Gata6-WT and KO^mye cells and observed a nearly complete inhibition of IL-1β secretion by the Gata6-KO^mye pMΦ (Fig 3A). This

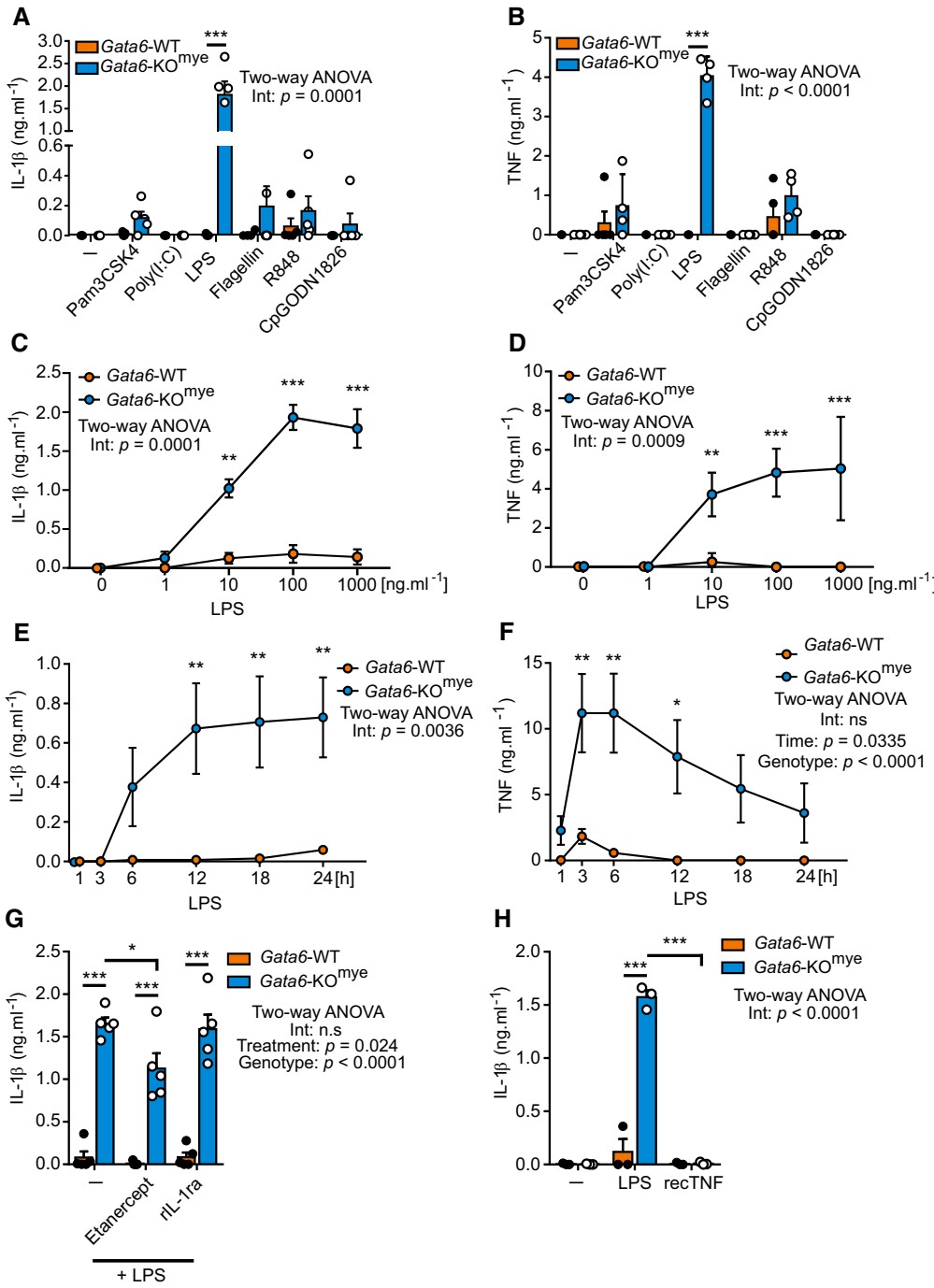

**Figure 1.**

◀

**Figure 1.  Aberrant cytokine release from LPS-stimulated Gata6-deficient resident peritoneal macrophages.**

A, B    Gata6-WT and Gata6-KO^mye peritoneal macrophages (pMΦ) were unstimulated (–) or stimulated with TLR2L Pam3CSK4 (500 ng/ml), TLR3L Poly(I:C) (1 μg/ml), TLR4L ultrapure LPS (100 ng/ml), TLR5L flagellin (100 ng/ml), TLR7 and 8L R848 (1 μg/ml) or TLR9L CpG ODN1826 (5 μM). Culture supernatants were collected 24 h after the start of stimulation and IL-1β and TNF ELISA were performed. *n* = 5, two-way ANOVA analysis with Tukey's multiple comparison post-test.

C, D    IL-1β (C) and TNF (D) ELISA of Gata6-WT and Gata6-KO^mye pMΦ stimulated for 24 h with the indicated LPS concentrations. *n* = 3, two-way ANOVA analysis with Sidak's multiple comparison post-test.

E, F    IL-1β (E) and TNF (F) ELISA from Gata6-WT and Gata6-KO^mye pMΦ stimulated with LPS (100 ng/ml) for the indicated times. *n* = 3, two-way ANOVA analysis with Sidak's multiple comparison post-test.

G       IL-1β ELISA of Gata6-WT and Gata6-KO^mye pMΦ stimulated for 18 h with 100 ng/ml LPS or recombinant IL-1 receptor antagonist (rIL-1ra), *n* = 4–5, two-way ANOVA analysis with Tukey's multiple comparison post-test.

H       IL-1β ELISA of Gata6-WT and Gata6-KO^mye pMΦ stimulated for 18 h with 100 ng/ml LPS or 100 ng/ml recombinant TNF (recTNF), *n* = 4–5, two-way ANOVA analysis with Tukey's multiple comparison post-test.

Data information: Data are expressed as mean ± SEM. *$P < 0.05$, **$P < 0.01$, ***$P < 0.001$.

observation suggested that the Gata6-WT cells actively inhibited IL-1β processing by the Gata6-KO^mye pMΦ. To determine whether this inhibition was due to direct contact between the cells or to a soluble factor secreted by the Gata6-WT pMΦ, we performed a Transwell experiment (Fig 3B). In this setting, the Gata6-WT pMΦ significantly inhibited IL-1β production from the Gata6-KO^mye pMΦ, however to a lesser extent than in the direct co-culture experiments. These data may suggest that the Gata6-WT pMΦ might be secreting a soluble molecule, albeit with a short half-life based on the reduced effect observed in the Transwell setting, inhibiting the pro-IL-1β processing pathway. Transcriptomic analysis of Gata6-WT and Gata6-KO^mye pMΦ (Rosas *et al*, 2014) showed alterations in many such potential candidate molecules (GEO: GSE47049); however, one of the greatest differentially expressed genes was prostaglandin I2 synthase (*Ptgis*) which converts prostaglandin H2 (PGH2) into prostacyclin (PGI2). Naïve Gata6-KO^mye pMΦ expressed significantly reduced amount of *Ptgis* mRNA (Fig 3C) and protein (Fig 3D) and significantly upregulated thromboxane A synthase 1 (*Tbxas1*) (Fig 3C), a direct competitor to Ptgis for the conversion of PGH2 into thromboxane A2 (TXA2). The expression of the two other enzymes implicated in the processing pathway of arachidonic acid (AA), cyclooxygenase 1 (*Ptgs1*) converting AA into prostaglandin G2 (PGG2) followed by PGH2 and prostaglandin E synthase 2 (*Ptges2*) processing PGH2 into prostaglandin E2 (PGE2) was less dramatically changed between Gata6-WT and Gata6-KO^mye pMΦ (Fig 3C). PGI2 has a very short half-life (< 2 min *in vivo*) and is rapidly hydrolysed to form 6-keto-prostaglandin F1α (6-keto-PGF1α), a metabolite that is readily detectable by mass spectrometry (Kunze & Vogt, 1971; Hamberg & Samuelsson, 1973; Jogee *et al*, 1983; Lewis & Dollery, 1983; Stitham *et al*, 2011). Therefore, to assess the impact of the Ptgis deficiency in Gata6-KO^mye cells on prostanoid production, we performed mass spectrometric analysis of the oxylipins in supernatants from Gata6-WT and KO^mye pMΦ cultured with or without 100 ng/ml LPS for 3 h. As expected, a significant reduction of 6-keto-PGF1α coupled with increased thromboxane B2 (TXB2) was observed from Gata6-KO^mye pMΦ after LPS stimulation (Fig 3E). Notably, prostaglandin E2 (PGE2) was also significantly decreased in Gata6-KO^mye pMΦ after LPS stimulation although the expression of the enzyme regulating its production (*Ptges2*) was unchanged (Fig 3C). These results confirmed an imbalanced prostanoid response in Gata6-KO^mye cells upon LPS stimulation (Fig 3F). It is important to note that both TXB2 and PGE2 were produced at much lower levels when compared to 6-keto-PGF1α, in

WT LPS-stimulated cells (approximately 10% of the levels). This suggests that PGI2 may normally be the dominant effector on downstream signalling in pMΦ (Norris *et al*, 2011). Altogether, these data suggest that Gata6-WT pMΦ are actively inhibiting the processing of IL-1β upon LPS stimulation and that a candidate for this effect may be PGI2 produced by the Gata6-dependent Ptgis enzyme.

## Prostacyclin inhibits IL-1β production via IL-10 induction in a Gata6-dependant pathway

PGI2 and LPS have both been previously shown to induce IL-10 production, including in MΦ (Fiorentino *et al*, 1991; Luttmann *et al*, 1999; Zhou *et al*, 2007). It is also known that pMΦ are predisposed to the production of IL-10 after stimulation with microbial products (Liao *et al*, 2016). Here, we observed that Gata6-KO^mye pMΦ produced significantly less IL-10, compared to Gata6-WT cells, after LPS stimulation (Fig 4A). To determine if this could be a consequence of reduced PGI2 levels, the impact of beraprost, cicaprost and iloprost (PGI2 analogues with various IP receptor affinities and specificities; Clapp & Gurung, 2015) on IL-10 generation was assessed (Fig 4B). When combined with LPS, all three analogues significantly increased IL-10 production in Gata6-KO^mye pMΦ (Fig 4B) compared to LPS and vehicle control (DMSO). In addition, all three analogues inhibited IL-1β production from Gata6-KO^mye pMΦ (Fig 4C). These data indicate that LPS-mediated induction of IL-10 in pMΦ is driven via a Gata6-PGI2-dependent pathway that, in turn, controls IL-1β production. As beraprost has been previously shown to have the most specific binding to IP receptor (Clapp & Gurung, 2015), we chose to conduct further experiments with this agonist only. Given the general imbalance in prostanoid content in Gata6-KO^mye pMΦ, we wondered if other prostanoids could also play a similar role in the control of IL-10 and IL-1β in pMΦ. To test this hypothesis, we stimulated the cells with beraprost, PGE2, U46619 (TXA2 receptor agonist) and picotamide (TXA2 receptor and Tbxas1 inhibitor) and analysed their IL-10 and IL-1β production (Fig 4D and E). PGE2 was able to induce IL-10 at a similar level to beraprost in Gata6-KO^mye pMΦ (Fig 4D) and also block IL-1β production (Fig 4E). However, neither U46619 nor picotamide affected IL-10 or IL-1β production in LPS-stimulated cells. These results indicate that prostanoid balance, especially the prevalence of PGE2 and PGI2 in pMΦ, controls the activation status of the cells and their response to inflammatory stimuli such as LPS. To investigate if the reduced amount of PGI2-derived IL-10 in Gata6-KO^mye

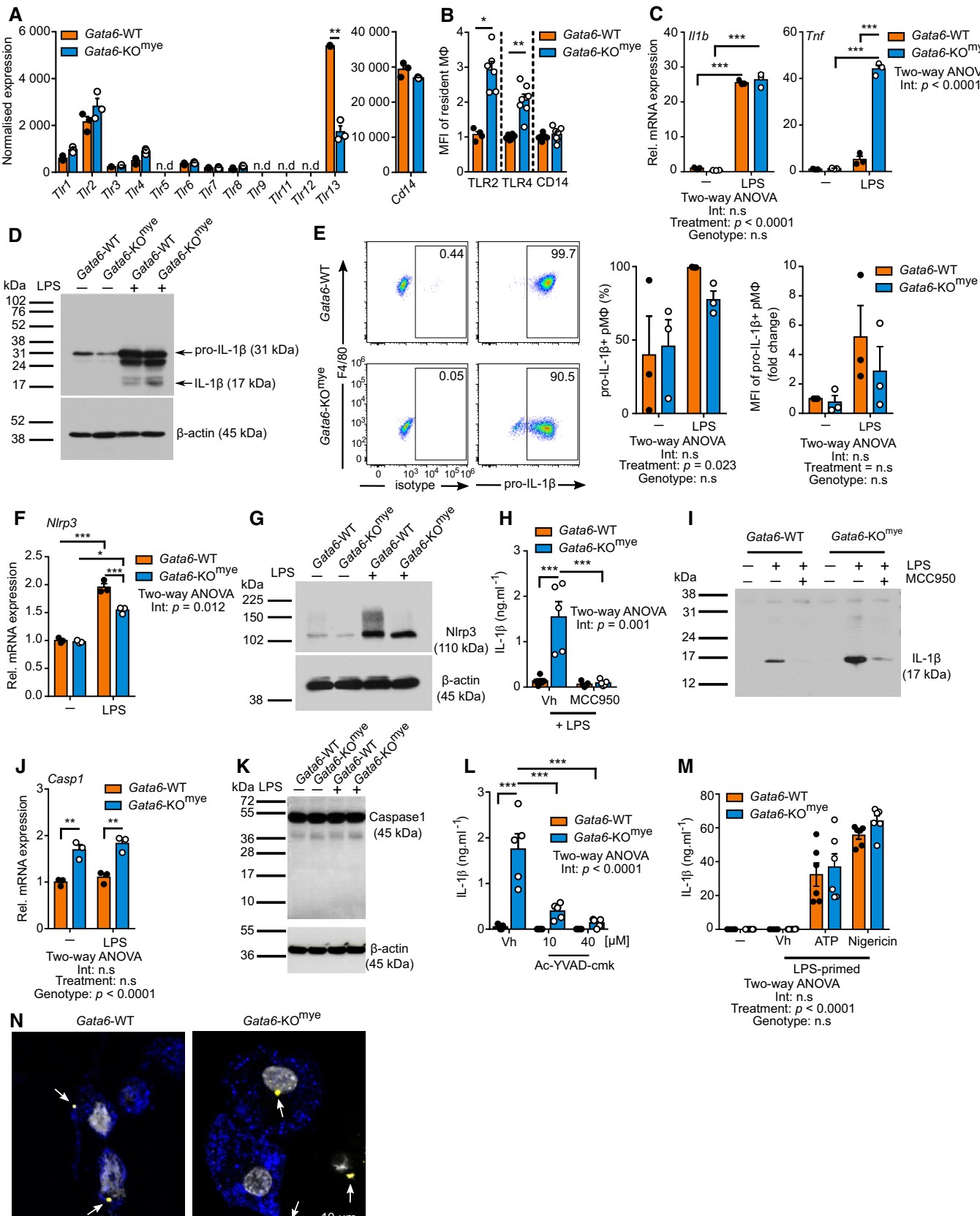

Figure 2.

**Figure 2. The aberrant release of IL-1β by Gata6-KO-deficient pMΦ follows classical inflammasome activation.**

A   Normalized expression of *Tlr* and *Cd14* genes following microarray analysis performed on unstimulated cell-sorted Gata6-WT and Gata6-KO^mye pMΦ. Data are shown as mean ± SEM from three biological replicates. Statistical significance was determined using empirical Bayesian statistic corrected for false discovery rate by the Benjamini–Hochberg procedure. n.d = non-detectable.

B   Mean fluorescence intensity (MFI) of extracellular TLR2, TLR4 and CD14 expression on naïve Gata6-WT and Gata6-KO^mye pMΦ. *n* = 4–7 individual mice per group.

C, D   *Il1b* and *Tnf* mRNA relative expression (C) and IL-1β Western blot protein analysis (D) of Gata6-WT and Gata6-KO^mye pMΦ stimulated with 100 ng/ml LPS for 3 and 6 h respectively. Data shown are representative of at least three independent experiments. Western blot was performed on whole cell lysates.

E   Representative dot plot, percentage and mean fluorescence intensity (MFI) analysis of pro-IL-1β^+ Gata6-WT and Gata6-KO^mye pMΦ flow cytometry analysis 3 h after stimulation with 100 ng/ml LPS. *n* = at least three independent experiments.

F   *Nlrp3* mRNA relative expression of Gata6-WT and Gata6-KO^mye pMΦ stimulated with 100 ng/ml LPS for 3 h. Data shown are pooled from three independent experiments.

G   Western blot protein analysis of Gata6-WT and Gata6-KO^mye pMΦ stimulated with 100 ng/ml LPS for 6 h. Data shown are representative of at least three independent experiments. Western blot was performed on whole cell lysates.

H, I   IL-1β ELISA (H) and Western blot protein analysis (I) of supernatants collected from Gata6-WT and Gata6-KO^mye pMΦ stimulated with 100 ng/ml LPS and either vehicle control (Vh, DMSO) or 10 μM MCC950 for 24 h (*n* = 5). Data shown in (H) are pooled from five independent replicates.

J, K   Caspase1 (*Casp1*) mRNA relative expression (J) and Western blot protein analysis (K) of Gata6-WT and Gata6-KO^mye pMΦ stimulated with 100 ng/ml LPS for 3 and 6 h respectively. Data shown are pooled from three independent experiments.

L   IL-1β ELISA of Gata6-WT and Gata6-KO^mye pMΦ stimulated with 100 ng/ml LPS and either vehicle control (Vh, DMSO) or Ac-YVAD-cmk for 24 h. Data shown are pooled of five independent replicates.

M   IL-1β ELISA of Gata6-WT and Gata6-KO^mye pMΦ stimulated with 100 ng/ml LPS for 3 h, followed by a 30 min pulse with either vehicle control (Vh), 5 mM ATP or 20 μM nigericin. Data shown are pooled of five independent replicates.

N   Representative picture of confocal immunofluorescence analysis of Gata6-WT and Gata6-KO^mye pMΦ stimulated with 100 ng/ml LPS for 3 h, followed by a 30 min pulse with 5 mM ATP. The white arrows show ASC specks. Scale bar = 10 μm.

Data information: Data are expressed as mean ± SEM and analysis were performed using two-way ANOVA analysis Tukey's multiple comparison post-test unless otherwise stated. *$P < 0.05$, **$P < 0.01$, ***$P < 0.001$.

pMΦ was responsible for their aberrant IL-1β production, we stimulated the cells with either beraprost or IL-10 together with LPS. Both beraprost and IL-10 significantly reduced the IL-1β production from Gata6-KO^mye cells after LPS stimulation, to levels that were comparable to those of the WT cells (Fig 4F). Lastly, we incubated the cells with an anti-IL-10 receptor (αIL-10R) neutralizing antibody, or an isotype-matched control, together with LPS and beraprost. As expected, the blockade of the IL-10 receptor, and thereby its signalling pathway, completely abrogated the effect of beraprost on IL-1β release (Fig 4G). Strikingly, IL-10R-inhibition, in the absence of any exogenous second signal, resulted in IL-1β production from Gata6-WT cells that was comparable to aberrant behaviour of Gata6-deficient counterparts. Together, these data demonstrate that WT cells actively block IL-1β production via a Gata6-PGI2-IL-10-dependent pathway, that is induced by LPS stimulation. To further understand the mechanism regulating this pathway, we performed a similar experiment, including the MCC950 inhibitor. When combined with αIL-10R treatment, MCC950 still inhibited the production of IL-1β after LPS stimulation, but only in Gata6-KO^mye pMΦ (Fig 4H), which coincided with a decrease in IL-10 production from these cells (Fig 4I). Notably, MCC950 had no effect on IL-1β and IL-10 production from Gata6-WT cells incubated with αIL-10R and LPS compared to αIL-10R and LPS-stimulated cells. These data suggest that the blockade of the IL-10 pathway in Gata6-WT pMΦ activates alternative pathways leading to increased stress and production of IL-1β in a non-canonical Nlrp3-independent pathway, in the absence of a secondary signal. As IL-10 had such a potent effect on IL-1β production in pMΦ after primary stimulation, we wondered if it could also affect IL-1β production in the presence of a secondary signal. We incubated Gata6-WT and Gata6-KO^mye pMΦ with LPS and IL-10 for 16 h, followed by a 30 min pulse with ATP. IL-10 was sufficient to significantly inhibit IL-1β production from both Gata6-WT and KO^mye cells (Fig 4J), consistent with its major role in the control of inflammation.

## Deficiency in Gata6 does not lead to dysfunctional mitochondria

Previous work has shown that IL-10 is essential to maintain mitochondria integrity in bone-marrow-derived macrophages (BMDM) (Ip *et al*, 2017). IL-10-deficient BMDM accumulate ROS which is known to activate NLRP3 inflammasome assembly leading to IL-1β production (Zhou *et al*, 2011). To investigate whether the aberrant IL-1β production we observed was mediated by mitochondrial dysfunction, we first analysed the mitochondrial membrane potential (MMP) and mitochondrial mass from Gata6-WT and KO^mye pMΦ *in vivo* by intraperitoneal injection of MitoTracker Green (MT green, total mitochondrial mass, independent of MMP) and Mito-Tracker Red (MT red, live mitochondria, dependent on MMP) to naïve mice before harvesting the peritoneal cells (Fig 5A). Both Gata6-WT and Gata6-KO^mye mitochondria showed similar mitochondrial mass characterized by a similar MT green staining but the Gata6-KO^mye pMΦ had significantly more MMP characterized by an increased MT red staining. We then investigated the effect of LPS, IL-10 and αIL-10R antibody on the mitochondria by performing *in vitro* experiments (Fig 5B). Unstimulated Gata6-KO^mye pMΦ had a significantly higher amount of mitochondrial mass and respiring mitochondria; however, addition of LPS reduced this phenotype to that of Gata6-WT pMΦ level. Inhibition of IL-10 signalling using αIL-10R resulted in lower mitochondrial mass of Gata6-WT pMΦ stimulated with LPS compared to Gata6-KO^mye cells. No significant difference in MMP could be observed between Gata6-WT and Gata6-KO^mye pMΦ (Fig 5B, right). Altogether, these data confirmed that deletion of Gata6 was not detrimental to mitochondria health, in naïve or LPS-stimulated pMΦ. In addition, staining with MitoSOX, a mitochondria-specific ROS indicator, showed that Gata6-WT and KO^mye pMΦ produced similar levels of superoxide (Fig 5C), regardless of the stimulation. To confirm that mitochondrial ROS were not implicated in the aberrant IL-1β production observed in the Gata6-KO^mye pMΦ, we treated the cells with the

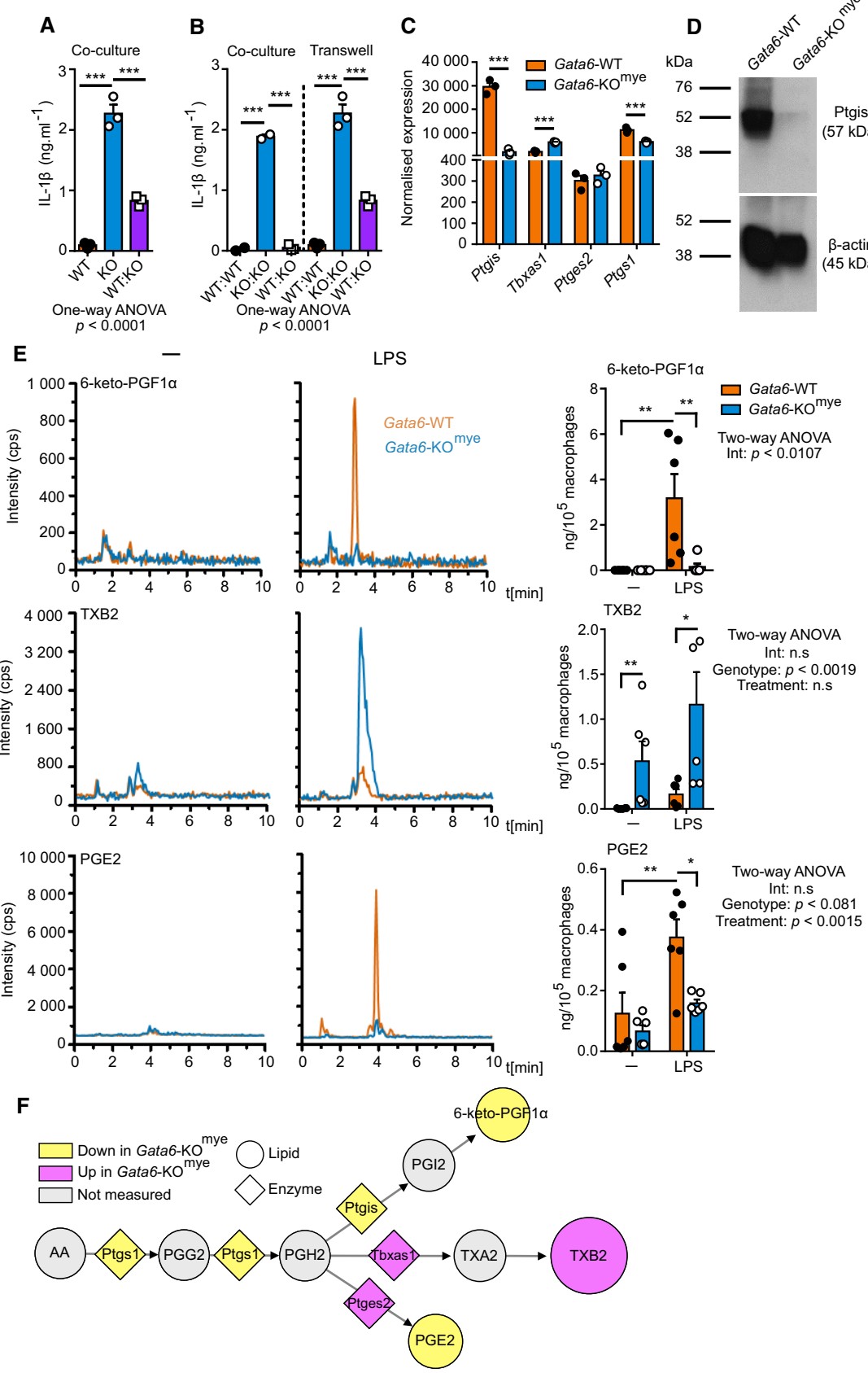

**Figure 3.**

◀

**Figure 3.  Prostanoid production is imbalanced in Gata6-KO^mye pMΦ.**

A   IL-1β ELISA analysis of supernatants of Gata6-WT and Gata6-KO^mye pMΦ in monoculture or co-cultured (ratio 1:1) and stimulated for 24 h with 100 ng/ml LPS. Data shown are pooled from three independent experiments. One-way ANOVA statistical analysis with Tukey's multiple comparison test was performed.

B   IL-1β ELISA analysis of supernatants of Gata6-WT and Gata6-KO^mye pMΦ co-cultured in the same well (co-culture) or using Transwell system (ratio 1:1) and stimulated for 24 h with 100 ng/ml LPS. Data shown are pooled from three independent experiments. One-way ANOVA statistical analysis with Tukey's multiple comparison test was performed.

C   Microarray analysis of *Ptgis*, *Tbxas1*, *Ptges2* and *Ptgs1* expression from Gata6-WT and Gata6-KO^mye pMΦ isolated from naïve mice. Data are shown as mean ± SEM from three biological replicates. Statistical significance was determined using empirical Bayesian statistic corrected for false discovery rate by the Benjamini–Hochberg procedure.

D   Western blot analysis of Ptgis protein level of unstimulated pMΦ from Gata6-WT and Gata6-KO^mye mice.

E   Mass spectrometry analysis of 6-keto-PGF1α, TXB2 and PGE2 content of Gata6-WT and Gata6-KO^mye pMΦ unstimulated (−) or stimulated for 3 h with 100 ng/ml LPS. *n* = 6. Two-way ANOVA statistical analysis with Tukey's multiple comparison post-test was performed.

F   Representation of the variation of the prostanoid synthesis pathway in Gata6-KO^mye pMΦ created using Cytoscape software. Circle shape represent lipids, diamond shape enzymes, yellow downregulation and purple upregulation of the expression/production in Gata6-KO^mye cells. The size of the circles represents relative levels observed in Gata6-KO^mye cells.

Data information: Data is shown as mean ± SEM. \*P < 0.05, \*\*P < 0.01, \*\*\*P < 0.001.

antioxidants MitoTEMPO, N-acetyl-L-cysteine (NAC) and mitoquinone (MitoQ) together with LPS and analysed their IL-1β production. All three antioxidants were unable to inhibit IL-1β production from Gata6-KO^mye cells (Fig 5D), confirming that mitochondrial ROS were not responsible for IL-1β production by Gata6-KO^mye pMΦ.

**PGI2-derived IL-10 controls IL-1β processing**

IL-10 has been reported to control the production of pro-inflammatory cytokines, such as TNF, by inhibiting their transcription (Murray, 2005). Here, to investigate how the PGI2-derived IL-10 regulates IL-1β in pMΦ in the absence of a secondary signal, we stimulated Gata6-WT and KO^mye pMΦ with beraprost, IL-10, αIL-10R or isotype control together with LPS for 3 h and analysed *Il1b* mRNA expression. IL-10 only slightly downregulated *Il1b* expression (Fig 6A) and αIL-10R increased it, most notably in Gata6-WT cells. In the presence of LPS, beraprost and IL-10 did not significantly affect *Nlrp3* or *Casp1* expression. However, when combined with LPS, αIL-10R significantly increased *Nlrp3* mRNA expression in both Gata6-WT and KO^mye pMΦ, compared to LPS and isotype-matched control. The treatments had no significant effect on *Casp1* expression, except for a small but significant downregulation induced by beraprost alone. Downstream analysis revealed that the protein level of the inflammasome components also remained quite stable following the various treatments (Fig 6B). LPS stimulation of Gata6-WT and KO^mye pMΦ for 6 h strongly induced pro-IL-1β and Nlrp3 protein level, and co-stimulation with IL-10 repeatedly decreased them. The various treatments did not dramatically affect caspase1 or ASC protein level, although it is interesting to note that ASC protein level showed strong variation between experiments. Caspase1 activity was evaluated using the FAM FLICA reagent on cells stimulated for 16 h. Gata6-KO^mye pMΦ showed a significant increase in caspase1 activity compared to Gata6-WT cells, independently of the stimulation (Fig 6C). LPS further increased caspase1 activity in Gata6-KO^mye pMΦ, and this could not be rescued by any other additional stimulation. Thorough analysis of pro-IL-1β by flow cytometry confirmed that LPS induced pro-IL-1β production in both Gata6-WT and KO^mye pMΦ and that co-stimulation with IL-10 decreased it in WT cells (Figs 6D and EV4B), in accordance with the mRNA and Western blot data. Blockade of the IL-10 pathway with αIL-10R increased pro-IL-1β, as observed on mRNA. Overall, these

data indicate a direct regulation of pro-IL-1β and Nlrp3 mRNA and protein synthesis via an IL-10-dependent pathway in pMΦ. It is interesting to note that a short time *in vitro* culture of pMΦ (16 h) was sufficient to strongly affect the baseline production of pro-IL-1β. Indeed, we could not detect any pro-IL-1β in freshly isolated and stained cells, compared to cells cultured unstimulated for 16 h (Figs 6D and EV4A and 5B). Despite all precautions taken to ensure optimal cell culture and minimizing activation of the cells, both Gata6-WT and Gata6-KO^mye pMΦ cultured for 16 h showed elevated pro-IL-1β content, compared to freshly isolated cells. This could, in part, explain the variation observed between samples and experiments. It is interesting to note that when challenged with LPS, Gata6-WT and KO^mye mice showed similar neutrophil recruitment to their peritoneal cavity (Fig EV5A) and increased pro-IL-1β content in pMΦ (Fig EV5B) as well as an increased IL-1β content in the peritoneal fluid, the latter of which was higher in the Gata6-KO^mye mice than the Gata6-WT mice (Fig EV5C). Although previous work have shown that IL-1β can directly modulate neutrophil recruitment (Martinon *et al*, 2006), it does not seem to be the case in our model. This discrepancy could possibly be explained by the fact that the defect we observe is localized to a single-cell population, which is reduced in number. Additionally, preliminary attempts to regulate neutrophil recruitment with anakinra had no effect (Fig EV5D). Further *in vitro* analyses showed that LPS upregulated *Il10* expression in Gata6-WT pMΦ but to a much lower extent in Gata6-KO^mye cells (Fig 6E). Interestingly, the addition of beraprost to LPS stimulation rescued *Il10* expression in Gata6-KO^mye cells, confirming the essential role of PGI2 in the induction of *Il10* in pMΦ after LPS stimulation. Direct stimulation with IL-10 was not able to rescue the *Il10* expression in Gata6-KO^mye pMΦ, suggesting that, in this setting, IL-10 does not control its own transcription. However, blocking IL-10 signalling using αIL-10R boosted *Il10* expression in Gata6-WT pMΦ stimulated with LPS, underlying the fact that IL-10 is an essential response to LPS stimulation and is at least partially dependent on PGI2. In contrast, *Tnf* expression in Gata6-KO^mye pMΦ after LPS stimulation was significantly downregulated by co-treatment with beraprost or IL-10 and αIL-10R antibody strongly induced *Tnf* expression in both Gata6-WT and Gata6-KO^mye cells (Fig 6E), indicating a direct transcriptional regulation of *Tnf* expression. Overall, these results show a clear differential regulation of classical pro-inflammatory cytokines production, such as TNF, and IL-1β production in pMΦ. PGI2-derived IL-10 at least partially

impairs the production of pro-IL-1β and Nlrp3 but not caspase1 nor ASC. Our data suggest that pMΦ possess an active inhibitory mechanism able to tightly control IL-1β processing in the absence of secondary signal. This inhibitory pathway not only regulates pro-IL-1β production but also maturation and possibly its subsequent processing (Fig 6F).

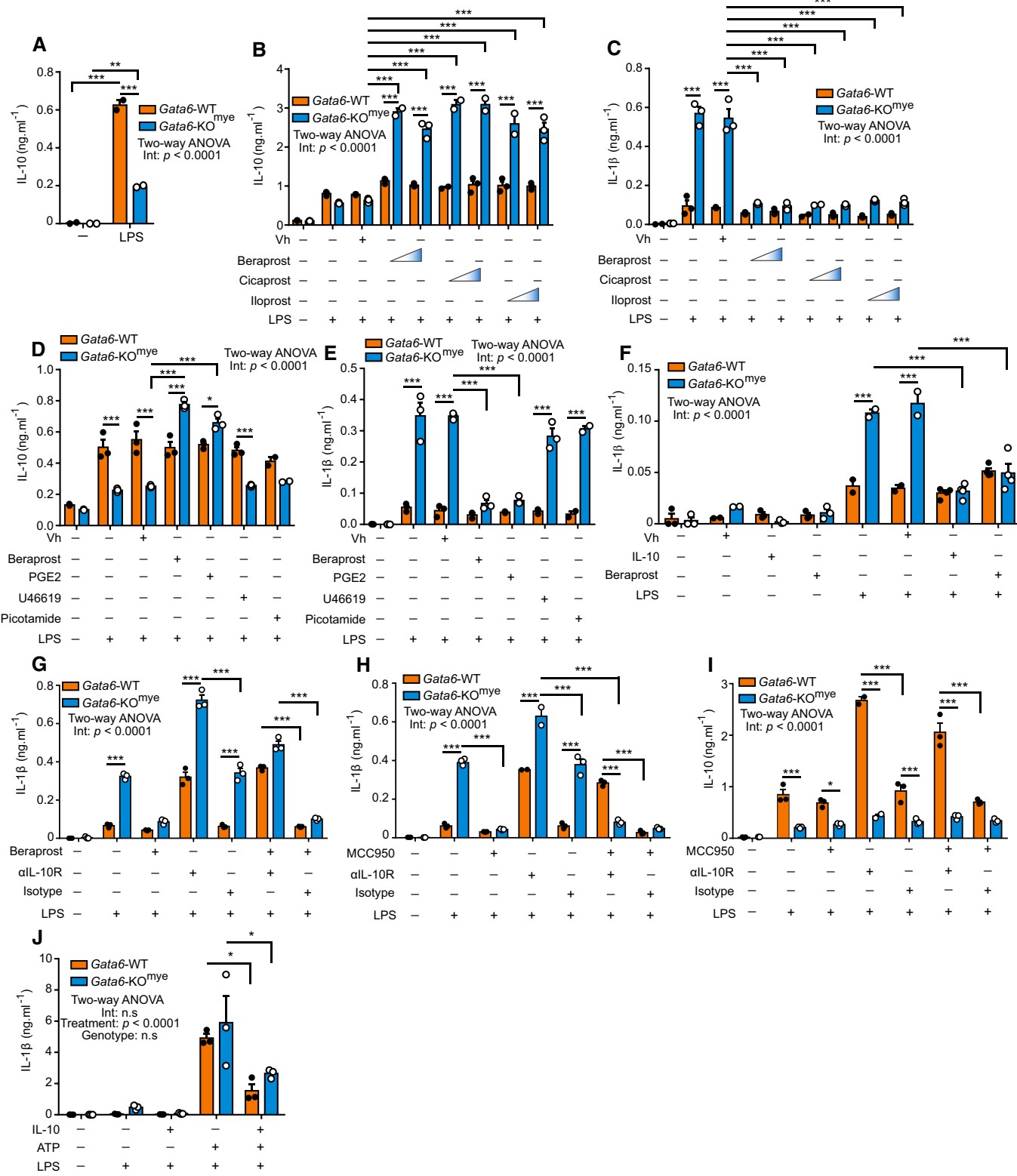

**Figure 4.**

Figure 4. The Gata6-PGI2-IL-10 pathway controls IL-1β production in Gata6-WT pMΦ.

A   IL-10 ELISA of supernatants of Gata6-WT and Gata6-KO<sup>mye</sup> pMΦ unstimulated (–) or stimulated with 100 ng/ml LPS. Data shown are representative of at least five independent experiments.

B, C   IL-10 (B) and IL-1β (C) ELISA of supernatants of Gata6-WT and Gata6-KO<sup>mye</sup> pMΦ unstimulated (–) or stimulated with 100 ng/ml LPS, beraprost (1 and 10 μM), cicaprost (1 and 10 μM), iloprost (1 or 10 nM) or vehicle control (Vh, DMSO). Data shown are representative of at least three independent experiments.

D, E   IL-10 (D) and IL-1β (E) ELISA of supernatants of Gata6-WT and Gata6-KO<sup>mye</sup> pMΦ unstimulated (–) or stimulated with 100 ng/ml LPS, 10 μM beraprost, 10 μM PGE2, 10 μM U46619, 100 μM picotamide or vehicle control (Vh, methyl acetate). Data shown are representative of at least three independent experiments.

F   IL-1β ELISA of supernatants of Gata6-WT and Gata6-KO<sup>mye</sup> pMΦ unstimulated (–) or stimulated with 100 ng/ml LPS, 10 ng/ml IL-10, 10 μM beraprost or vehicle control (Vh). Data shown are representative of at least 3 independent experiments.

G   IL-1β ELISA of supernatants of Gata6-WT and Gata6-KO<sup>mye</sup> pMΦ unstimulated (–) or stimulated with 100 ng/ml LPS, 10 μM beraprost, 5 μg/ml αIL-10R or isotype antibody. Data shown are representative of at least three independent experiments.

H, I   IL-1β (H) and IL-10 (I) ELISA of supernatants of Gata6-WT and Gata6-KO<sup>mye</sup> pMΦ unstimulated (–) or stimulated with 100 ng/ml LPS, 10 μM MCC950, 5 μg/ml αIL-10R or isotype antibody All stimulations were performed for 16 h. Data shown are representative of at least three independent experiments.

J   IL-1β ELISA of supernatants of Gata6-WT and Gata6-KO<sup>mye</sup> pMΦ unstimulated (–) or stimulated with 100 ng/ml LPS, 10 ng/ml IL-10 and 5 mM ATP. Data shown are a pool of three independent experiments.

Data information: Data are expressed as mean ± SEM and two-way ANOVA statistical analysis with Tukey's multiple comparison post-test were performed. *$P < 0.05$, **$P < 0.01$, ***$P < 0.001$.

## Discussion

Our study here shows that, in the absence of secondary signal, WT tissue-resident pMΦ actively block IL-1β processing via a Gata6-PGI2-IL-10-dependent pathway. The absence of this inhibitory pathway leads to a significant production of IL-1β, as observed in the Gata6-KO<sup>mye</sup> tissue-resident pMΦ or when using αIL-10R antibody in Gata6-WT pMΦ. IL-1β production and secretion is a complex mechanism, still partially uncharacterized. Upon LPS stimulation, the immature pro-IL-1β form accumulates and its processing is restricted, unless permitted by a secondary signal. The role of IL-1β as an alarmin makes it an extremely potent cytokine, whose production must be tightly controlled to avoid unnecessary inflammatory damage to the surrounding area. The concept of specific mechanisms restraining activation of the inflammasome in the absence of secondary signal therefore seems essential. Recent work showed that tissue MΦ actively attempt to suppress inflammation and that pMΦ promote local injury repair (Wang & Kubes, 2016; Uderhardt et al, 2019). Our work provides a mechanism for constraint of pro-inflammatory responses at the cellular level, both autocrine and paracrine, supporting the idea that upon pro-inflammatory signal recognition, tissue-resident pMΦ are programmed to first control and dampen inflammation. In the presence of a secondary insult, such as in our case inflammasome-activating molecules, this first controlling response may be over-ruled and pMΦ would then actively promote inflammation by the release of alarmins such as IL-1β and recruitment of neutrophils (Martinon et al, 2006).

The fact that the active inhibitory mechanism we describe here has not been previously reported is likely explained by the lack of tools to study IL-1β regulation. Although genetic defects leading to spontaneous IL-1β secretion by MΦ have been reported (Saitoh et al, 2008; Vince et al, 2012; Duong et al, 2015), its maturation and release are usually investigated in the presence of highly potent secondary signals such as ATP and nigericin.

Previous work (Ip et al, 2017) has shown that deficiency in IL-10 in BMDM leads to dysfunctional mitochondria releasing mitochondrial ROS which act as a secondary signal, activating the inflammasome and leading to the aberrant release of IL-1β. Our results clearly show that although Gata6-KO<sup>mye</sup> pMΦ produce a much lower amount of PGI2-derived IL-10 compared to Gata6-WT pMΦ, their mitochondria remain healthy and functional upon LPS

stimulation. This discrepancy can first be explained by the fact that different types of MΦ have been used (BMDM versus tissue-resident pMΦ). Previous work (Norris et al, 2011) showed that BMDM and pMΦ differently utilize the enzymes implicated in the AA pathway and thereby produce differential amount of prostanoids. Upon TLR4 activation, pMΦ synthesize PGI2, among others, while BMDM do not, due to the absence, or really low expression of Ptgis. Our results clearly show a strong preference for PGI2 production in pMΦ, approximately 10-fold higher levels of PGI2 are produced than PGE2. When comparing tissue-resident pMΦ, BMDM, thioglycolate-induced inflammatory peritoneal MΦ and RAW 264.7 MΦ cell line, the authors showed that tissue-resident pMΦ had the highest expression of Ptgis. Altogether, these data suggest that depending on their origin, MΦ differentially respond to TLR4 stimulation by favouring the production of specific prostanoid species. Their response to TLR4 ligand should therefore not be directly compared, at least from a prostanoid synthesis point of view. Next, it is important to note that the Gata6-KO<sup>mye</sup> pMΦ are still able to produce some IL-10 (about 3–4 times less than Gata6-WT pMΦ) after LPS stimulation, whereas Ip and colleagues (Ip et al, 2017) used IL-10-deficient MΦ. IL-10 is known to play an important role in homeostasis (Shouval et al, 2014; Zigmond et al, 2014; Girard-Madoux et al, 2016), inflammation regulation (Mosser & Zhang, 2008) and tissue regeneration (Siqueira Mietto et al, 2015) and a complete loss of IL-10 could explain cell instability and mitochondria dysfunction. The amount of IL-10 produced must be finely tuned, and although a small amount seems to be sufficient for tissue-resident pMΦ mitochondria health, it is less effective for IL-1β regulation.

It is interesting to note that blockade of the IL-10 signalling pathway using αIL-10R antibody induced both mRNA and protein expression of IL-1β and Nlrp3, suggesting that a complete blockade of the signalling pathway directly affects IL-1β and inflammasome components production and that the small amount of IL-10 still produced in Gata6-KO<sup>mye</sup> pMΦ is sufficient to avoid this direct impact.

We also show that PGI2-derived IL-10 inhibits the production of both classically produced cytokines such as TNF and IL-1β but via two completely different mechanisms, TNF being inhibited directly at the mRNA level and IL-1β during a later stage of processing. These data point out that the anti-inflammatory effect of IL-10 is dependent on its target and emphasizes the fact that careful

consideration should be taken when modulating the response to inflammation.

Marketed therapeutics targeting IL-1β are mostly focused on blocking its receptor (Dinarello *et al*, 2012) and thereby its signalling. Although proven efficient in reducing symptoms in some contexts (Cunnane *et al*, 2001; Bresnihan *et al*, 2004; Goldbach-

Mansky *et al*, 2006; Ridker *et al*, 2017), these drugs do not treat the cause of aberrant IL-1β production and need to be constantly taken to prevent overwhelming inflammation. Our work provides insight into IL-1β regulation, which will be necessary to provide new therapeutic options directed to the source of the dysfunction and improve IL-1β control to ameliorate diseases.

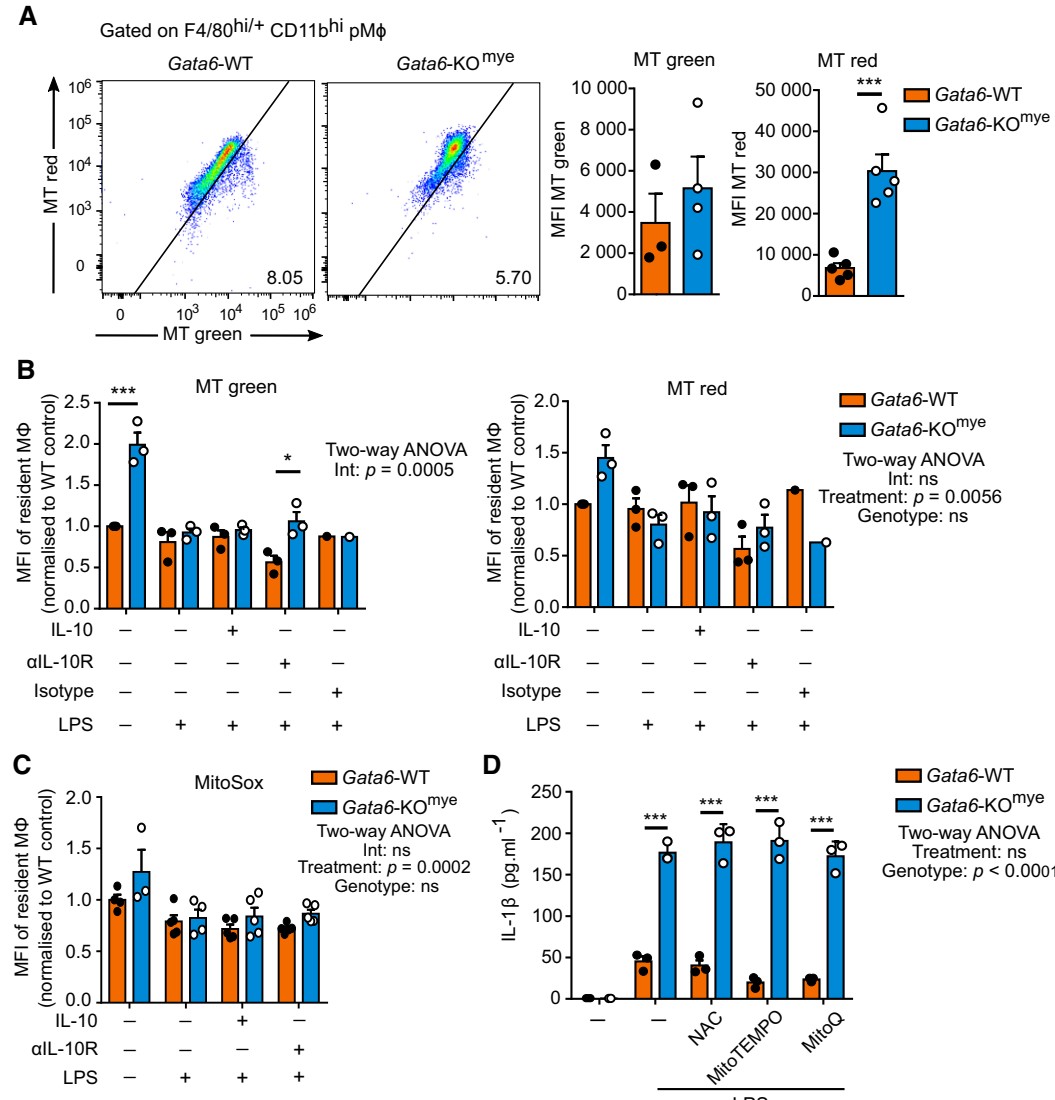

**Figure 5.  The aberrant IL-1β production by Gata6-KO^mye pMΦ is not led by mitochondria dysfunction.**

A   Flow cytometry analysis of MitoTracker (MT) Green and Red integration in Gata6-WT and Gata6-KO^mye naïve pMΦ (gated on F4/80^hi/+ Tim4+) *in vivo*, 30 min after intraperitoneal injection (i.p.) of 1 μM of MitoTracker Green and Red. Data shown are representative of 3–5 independent mice of each genotype and are expressed as mean ± SEM. Student's *t*-test analysis was performed.

B   Flow cytometry analysis of MitoTracker Green and Red staining in Gata6-WT and Gata6-KO^mye pMΦ (gated on F4/80^hi/+ Tim4+) unstimulated (−) or after LPS (100 ng/ml), IL-10 (10 ng/ml), αIL-10R (5 μg/ml) or isotype (5 μg/ml) stimulation for 16 h *in vitro*. Data shown are pooled from three independent experiments and normalized to WT unstimulated.

C   Flow cytometry analysis of MitoSOX staining in Gata6-WT and Gata6-KO^mye pMΦ (gated on F4/80^hi/+ Tim4+) unstimulated (−) or after LPS (100 ng/ml), IL-10 (10 ng/ml) and αIL-10R (5 μg/ml) stimulation for 16 h *in vitro*. Data shown are pooled from 2 independent experiments and normalized to WT unstimulated.

D   IL-1β ELISA of supernatants of Gata6-WT and Gata6-KO^mye pMΦ unstimulated (−) or stimulated with 100 ng/ml LPS, 500 μM MitoTEMPO, 10 mM NAC, 0.5 μM MitoQ for 16 h. Data are representative of at least 3 experiments.

Data information: Data are expressed as mean ± SEM, and two-way ANOVA statistical analysis with Tukey's multiple comparison post-test was performed unless otherwise stated. *P < 0.05, ***P < 0.001.

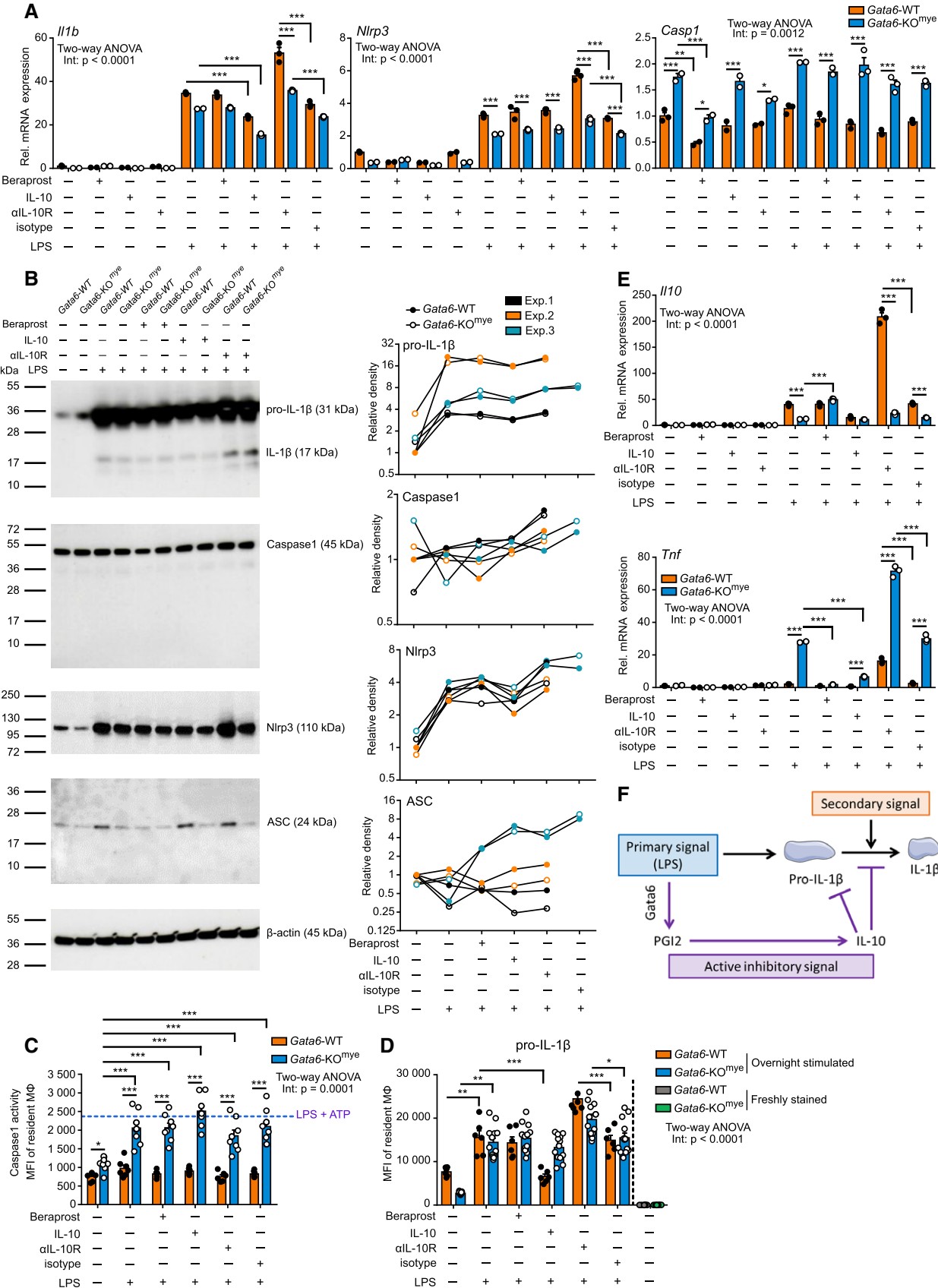

**Figure 6.**

**Figure 6.  PGI2-derived IL-10 controls IL-1β processing.**

A   mRNA expression analysis of *Il1b*, *Nlrp3* and *caspase1* (*Casp1*) of Gata6-WT and Gata6-KO[mye] pMΦ stimulated for 3 h with 100 ng/ml LPS, 10 μM beraprost, 10 ng/ml IL-10, 5 μg/ml αIL-10R or 5 μg/ml isotype. Data are representative of at least three independent experiments.

B   Western blot analysis (left) and quantification (right) of Gata6-WT and Gata6-KO[mye] pMΦ stimulated for 6 h with 100 ng/ml LPS, 10 μM beraprost, 10 ng/ml IL-10 or 5 μg/ml αIL-10R. Data are representative of three independent experiments.

C   Caspase1 activity analysis of Gata6-WT and Gata6-KO[mye] pMΦ stimulated for 16 h with 100 ng/ml LPS, 10 μM beraprost, 10 ng/ml IL-10, 5 μg/ml αIL-10R or 5 μg/ml isotype. *n* = 5–8 individual mice.

D   Mean fluorescence intensity (MFI) of pro-IL-1β of Gata6-WT and Gata6-KO[mye] pMΦ stimulated for 3 h with 100 ng/ml LPS, 10 μM beraprost, 10 ng/ml IL-10, 5 μg/ml αIL-10R or 5 μg/ml isotype for 16 h or freshly isolated. *n* = 6–13 individual mice. Data were log-transformed before performing statistical analysis.

E   mRNA expression analysis of *Il10* and *Tnf* of Gata6-WT and Gata6-KO[mye] pMΦ stimulated for 3 h with 100 ng/ml LPS, 10 μM beraprost, 10 ng/ml IL-10 or 5 μg/ml αIL-10R. Data are representative of at least three independent experiments.

F   Representation of the mechanism of action of the Gata6-PGI2-IL-10 active inhibitory signal on IL-1β processing.

Data information: Data are expressed as mean ± SEM, and two-way ANOVA statistical analysis with Tukey's multiple comparison post-test was performed. *$P < 0.05$, **$P < 0.01$, ***$P < 0.001$.

## Materials and Methods

### Mice

Lysozyme M Cre-recombinase "knock-in" congenic mice on the C57BL/6 background ('*Lyz2^Cre*', B6.129P2-*Lyz2^{tm1(cre)Ifo/J}*) and conditional "floxed" Gata6-deficient mice ('*Gata6^Fl*', *Gata6^{tm2.1Sad/J}*) were obtained from the Jackson Labs and bred in our animal facilities as previously described (Rosas *et al*, 2014). All mice were sex-matched and between 6–12 weeks of age at the time of use, unless otherwise stated. All animal work was conducted in accordance with Institutional and UK Home Office guidelines.

### Reagents

Ultra-pure LPS-EB from *E. coli* O111:B4, Pam3CSK4, Poly(I:C) (HMW), Flagellin, R848, and CpG ODN 1826 were purchased from InvivoGen. MitoTEMPO, *N*-acetyl-L-cysteine, Ac-YVAD-cmk, nigericin, ATP and picotamide were purchased from Sigma. Beraprost, cicaprost, iloprost, U46619 and mitoquinone were purchased from Cambridge Bioscience. PGE2 was purchased from R&D. MCC950 sodium salt was a kind gift from Prof. Avril Robertson (University of Queensland). Recombinant IL-1ra was purchased from R&D System. Recombinant IL-10 and TNF were purchased from PeproTech and the anti-IL-10R Antibody (clone 1B1.3A, Bio X Cell). Etanercept was purchased from Sigma-Aldrich. GolgiBlock was purchased from BD Biosciences.

### Resident peritoneal macrophages (pMΦs) isolation and culture

Resident peritoneal macrophages (pMΦs) were obtained via peritoneal lavage with 5 ml lavage solution (PBS (Invitrogen) supplemented with 5 mM EDTA and 4% foetal calf serum (FCS)). Lavages of the same genotype were pooled and resuspended in complete medium (RPMI 1640 supplemented with 10% FCS, 100 U/ml penicillin, 10 μg/ml streptomycin and 400 μM L-glutamine (Invitrogen)). Typically, the cells were plated and left to adhere for 3 h at 37°C, 5% $CO_2$ before being washed two times with warm complete medium and further stimulated as indicated. When needed, cells were plated on Transwell permeable supports (Corning).

### Flow cytometry

The following antibodies were used for flow cytometry analysis: CD11b (clone M1/70) was purchased from BD Biosciences; MHCII (clone M5/114.15.2), Tim4 (clone RMT4-54), F4/80 (clone BM8), Ly6G (clone 1A8) from BioLegend; pro-IL-1β (clone NJTEN3) and matching isotype control antibody were purchased from eBioscience. For extracellular staining, cells were collected and resuspended in flow cytometry buffer (4% FCS in PBS) containing 4 μg/ml of Fc receptor blocking antibody 2.4G2 (homemade) for 15 min on ice. Cells were then stained for 30 min with the indicated antibodies. For intracellular staining, cells were first fixed for 15 min with 2% paraformaldehyde (PFA) and then permeabilized and blocked at 4°C for 30 min in permeabilization buffer (0.5% bovine serum albumin (BSA), 5 mM EDTA, 2 mM $NaN_3$ and 0.5% saponin) containing 4 μg/ml 2.4G2 blocking antibody (homemade). Cells were then stained with the indicated antibodies for 1 h at 4°C in permeabilization buffer. Flow cytometry was performed on Cyan (Beckman Coulter) or Attune (Thermo Fisher) flow cytometer and analysed with FlowJo software.

### Cytokine measurement

Supernatants were collected at the indicated time points, transferred to V-bottom 96-well plate and centrifuged at 500 ×*g* for 5 min. The supernatant was carefully removed and placed into a fresh 96-well V-bottom plate (cell-free supernatant). If the supernatants were not assessed for cytokine production immediately, samples were stored at −80°C until use. Supernatants were assayed for IL-1β (R&D Systems), TNF and IL-10 (BD OptEIA™, BD Biosciences) following the manufacturer's instructions.

### Caspase1 activity measurement

Caspase1 activity was evaluated using the FAM FLICA™ Caspase-1 Kit (Bio-Rad), following the manufacturer's instructions.

### Mitochondrial content measurement

MitoTracker Green FM, MitoTracker DeepRed and MitoSOX Red mitochondrial superoxide indicator were purchased from Thermo Fisher. *In vitro* staining was performed following the manufacturer's instructions. For *in vivo* staining, naïve mice were intraperitoneally (i.p.) injected with 1 μM MitoTracker Green FM or MitoTracker DeepRed diluted in PBS. Mice were sacrificed 30 min after injection, and peritoneal cells were collected as described above. The cells were then stained for cell surface antigens and subsequently analysed by flow cytometry.

## Immunoblotting

Proteins were isolated using RIPA buffer (Santa Cruz), following manufacturer's instructions. Cell-free culture supernatants were concentrated using methanol/chloroform precipitation. Briefly, one volume of methanol and one-quarter volume of chloroform were added to the supernatant, vortexed vigorously for 20 s and then centrifuged at $20,000 \times g$ for 10 min. After centrifugation, the upper phase was aspirated and one volume of methanol was added, vortexed for 20 s and centrifuged at $20,000 \times g$ for 5 min. The supernatant was removed, and the pellets briefly dried in a heat block set at 55°C for 10 min. After drying, the pellet was resuspended in $1 \times$ laemmli buffer (Bio-Rad), vortexed and heated to 95°C for 5 min. Proteins were run on SDS–PAGE gels and transferred to PVDF membranes (Bio-Rad). The primary antibodies used were goat anti-mouse IL-1β (catalogue number AF-401-NA; R&D Systems), rat anti-human/mouse Nlrp3 (clone 768319; R&D Systems), mouse anti-mouse caspase1 p20 (clone Casper-1; Adipogen), rabbit anti-mouse ASC (clone AL177, Adipogen), rabbit anti-mouse Ptgis (catalogue number ab23668, Abcam) and mouse anti-mouse β-actin (clone AC-74; Sigma). The secondary antibodies used were rabbit anti-goat (Dako), goat anti-rabbit (Dako), sheep anti-mouse (GE healthcare life sciences) and rabbit anti-rat (GE healthcare life sciences). Western blots were quantified using ImageJ.

## Quantitative real-time PCR

Total RNA was isolated from cells using RNeasy Mini or Micro Kit (Qiagen) following the manufacturer's instructions, and at least 350 ng were reverse transcribed using the High-Capacity cDNA Reverse Transcription Kit (Applied Biosystems). mRNA levels were quantified using a ViiA™ 7 Real-Time PCR System (Applied Biosystem) and Power SYBR Green PCR Master Mix (Thermo Fisher). The gene expression values were normalized to *Ywhaz* expression and normalized to WT control. The primers used were as followed: *Il1b* 5′-ATGAAGGGCTGCTTCCAAAC-3′ and 5′-ATGTGCTGCTGCGAGA TTTG-3′; *Nlrp3*, 5′-TGGGCAACAATGATCTTGGC-3′ and 5′-TTTC ACCCAACTGTAGGCTCTG-3′; *Tnf*, 5′-TAGCCCACGTCGTAGCAAA C-3′ and 5′-ACAAGGTACAACCCATCGGC-3′; *Casp1*, 5′-CACAGCTC TGGAGATGGTGA-3′ and 5′-CTTTCAAGCTTGGGCACTTC-3′; *Il10*, 5′-GGTTGCCAAGCCTTATCGGA-3′ and 5′-GAGAAATCGATGACAG CGCC-3′; *Ywhaz* 5′-TTGAGCAGAAGACGGAAGGT-3′ and 5′-GAA GCATTGGGGATCAAGAA-3′.

## Lipid analysis

Peritoneal lavages of Gata6-WT and Gata6-KO^mye mice were collected. Cells were washed with complete medium and plated for 3 h with or without 100 ng/ml LPS. Supernatants were collected, snap frozen and stored at −80°C prior to lipid extraction and analysis. Lipids were extracted by adding a 1.25 ml solvent mixture (1 M acetic acid/isopropanol/hexane; 2:20:30, v/v/v) to 0.5 ml supernatants in a glass extraction vial and vortexed for 30 s. 1.25 ml hexane was added to samples and after vortexing for 30 sec., tubes were centrifuged (500 *g* for 5 min at 4°C) to recover lipids in the upper hexane layer (aqueous phase), which was transferred to a clean tube. Aqueous samples were re-extracted as above by addition of 1.25 ml hexane. The combined hexane layers were dried in a

RapidVap (Labconco) at 30°C, resuspended in 100 μl methanol and stored at −80°C. Lipids were then separated on a C18 Spherisorb ODS2, 5 μm, 150 × 4.6-mm column (Waters) using a gradient of 50–90% B over 20 min (A, water:acetonitrile:acetic acid, 75:25:0.1; B, methanol:acetonitrile:acetic acid, 60:40:0.1) with a flow rate of 1 ml/min. Products were quantitated by LC/MS/MS electrospray ionization on an Applied Biosystems 4000 Q-Trap using parent-to-daughter transitions of $m/z$ 351.2–$m/z$ 271.1 (PGE2), $m/z$ 355.2–$m/z$ 275.1 (PGE2-d4), $m/z$ 369.2–$m/z$ 169.1 (TXB2), $m/z$ 369.2–$m/z$ 245.1 (6-keto-PGF1a), all [M-H], with collision energies of −20 to −36 V. Source parameters: TEM 650, IS -4500, CUR -35, GS1 60, GS2 30, EP -10. Products were identified and quantified using PGE2, 6-Keto-PGF1α, TXB2 and PGE2-d4 (10 ng each was added to samples prior to extraction) standards run in parallel under the same conditions. Chromatographic peaks were integrated using Analyst software (Sciex). The criteria for assigning a peak were signal:noise of at least 3:1 and with at least 7 points across a peak. The ratio of analyte peak areas to internal standard was taken and lipids quantified using a standard curve made up and run at the same time as the samples and the amount of lipids were normalized to the number of pMΦ.

## Immunofluorescence

Cells were plated in complete medium on cover glass (VWR, thickness no. 1) and left to adhere for 3 h before washing 3 times with complete medium. Following indicated treatment, cells were fixed for 15 min with 4% PFA and permeabilized and blocked for 30 min with PBS containing 0.1% Triton X-100 (Sigma), 10% FCS and 10% rabbit serum. The cells were then stained overnight with the following antibody: rabbit anti-mouse ASC (catalogue number AL177, Adipogen) and F4/80-AF488 (clone BM8, BioLegend). Following washes, secondary antibody (Goat anti-rabbit-AF594, Thermo Fisher) was applied for 1 h. DAPI (Thermo Fisher) was applied as counterstain, and cells were mounted on microscope slides (Thermo Scientific) using fluorescent mounting medium (Dako). Pictures were taken using LSM800 confocal laser scanning microscope (Zeiss).

## Statistical analysis

Results are expressed as the mean ± SEM. Data were analysed with two-way ANOVA followed by post-tests, unless otherwise stated. A $P < 0.05$ was considered statistically significant (*$P < 0.05$, **$P < 0.01$, ***$P < 0.001$). All statistics were performed using GraphPad Prism 6 software.

# Data availability

Microarray expression data from wild-type and Gata6-deficient tissue-resident peritoneal macrophages data are available via GEO (https://www.ncbi.nlm.nih.gov/geo/) (GSE47049).

**Expanded View** for this article is available online.

## Acknowledgements

We thank the staff of our animal facilities for the care of the animals. P.R.T is funded by the Wellcome Trust Investigator Award (107964/Z/15/Z) and the UK

Dementia Research Institute and L.C.D was funded by the Wellcome Trust (103973/Z/14/Z). C.E.B is funded by the Wellcome Trust Investigator Award (108045/Z/15/Z). S.J.O was funded by a Sir Henry Dale Fellowship jointly funded by the Wellcome Trust and the Royal Society (Grant Number 099953/Z/12/Z). We would like to thank Drs. Mathew Clement and Gareth Jones for assistance with reagents.

## Author contributions

Conceptualization: NI, RJP, MR, LCD, CEB, PRT; Methodology: NI, RJP, MR, SJO; Experimentation: NI, RJP, MR, MAC, DF, VJT; Resources: SJO, AABR, VO'D, PRT; Data analysis: NI, RJP, MR, CEB, VO'D, PRT; Writing & Editing: NI, PRT; Supervision: PRT All authors read and commented on the manuscript.

## Conflict of interest

AABR is a co-inventor on granted patents (US 10,538,487, EP 3259253) and patent applications (WO2018215818, WO2017140778, WO2016131098) for NLRP3 inhibitors, which are licensed to Inflazome Ltd, a company headquartered in Dublin, Ireland. Inflazome is developing drugs that target the NLRP3 inflammasome to address unmet clinical needs in inflammatory disease.

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
