## [Review Process File · The EMBO Journal]

Tissue resident macrophages actively suppress $\text{IL-1}\beta$ release via a reactive prostanoid/ IL-10 pathway

NATACHA IPSEIZ, Robert Pickering, Marcela Rosas, Victoria Tyrrell, Luke Davies, Selinda Orr, Magdalena Czubala, Dina Fathalla, Avril Robertson, Clare Bryant, Valerie O'Donnell, and Philip Taylor
DOI: [10.15252/embj.2019103454](https://doi.org/10.15252/embj.2019103454)

Corresponding author(s): Philip Taylor (taylorpr@cardiff.ac.uk)

Review Timeline:

Submission Date:	13th Sep 19
Editorial Decision:	14th Oct 19
Revision Received:	12th Mar 20
Editorial Decision:	3rd Apr 20
Revision Received:	17th Apr 20
Editorial Decision:	27th Apr 20
Revision Received:	28th Apr 20
Accepted:	30th Apr 20

Editor: Karin Dumstrei

Transaction Report:

Dear Dr. Taylor,

Thank you for submitting your manuscript to The EMBO Journal. Your study has now been seen by two referees and their comments are provided below.

As you can see from the comments, the referees find the analysis interesting and suitable for publication here. They raise a number of good issues that should be resolved in order to consider publication here. Should you be able to address the comments raised in full then I would like you to invite you to submit a revised version. I should add that it is EMBO Journal policy to allow only a single major round of revision, and that it is therefore important to address the raised issues at this stage.

Let me know if we need to discuss the revisions further - happy to do so

When preparing your letter of response to the referees' comments, please bear in mind that this will form part of the Review Process File, and will therefore be available online to the community. For more details on our Transparent Editorial Process, please visit our website:

<https://www.embopress.org/page/journal/14602075/authorguide#transparentprocess>

Thank you for the opportunity to consider your work for publication. I look forward to your revision.

Yours sincerely,

Karin Dumstrei, PhD
Senior Editor
The EMBO Journal

- a point-by-point response to the referees' comments, with a detailed description of the changes made (as a word file).

- a word file of the manuscript text.

- individual production quality figure files (one file per figure)

- a complete author checklist, which you can download from our author guidelines (<https://www.embopress.org/page/journal/14602075/authorguide>).

- Expanded View files (replacing Supplementary Information)

Further information is available in our Guide For Authors:

The revision must be submitted online within 90 days; please click on the link below to submit the revision online before 12th Jan 2020.

Link Not Available

Referee #1:

This is a very interesting study identifying a novel anti-inflammatory pathway in large Gata6 peritoneal macrophage. The mechanism involves PGI2 and IL10. I have a few questions to help give this in vivo relevance.

1) In the peritoneum there are large and small peritoneal cavity macrophage. Do the large regulate the small? If so how. Does the PGI2 and IL10 directly affect monocyte derived macrophage recruitment, cytokine production etc.

2) What happens in GATA6 deficient mice with LPS stimulation IP. This would help to elucidate the importance of this pathway in vivo. Are there alterations in neutrophil influx?

3) Infectious agents tend to make these large peritoneal macrophage disappear. Although we still struggle to understand this macrophage disappearance reaction could the authors comment what happens after LPS delivery to the macrophage and cytokine production?

4) This sentence in the discussion and the reference are not correct. "Recently, Uderhardt and colleagues showed that pMΦ actively attempt to suppress inflammation and initiate local injury repair (26)." In this paper Germain and colleagues were not looking at the GATA6+ peritoneal macrophages but rather the tissue macrophages in the abdominal wall. Moreover, they simply show that the macrophages cloak and there was no repair function since they were only looking at the death of a single cell. The repair function of peritoneal macrophages was shown 3 years earlier by

Referee #2:

The study by Ipseiz et al describes a new GATA6-induced pathway in resident peritoneal macrophages that limits IL-1b release following TLR activation. The authors provide evidence that GATA-6 deficient macrophages release excessive amount of IL-1b compared to WT cells due to enhanced processing of pro-IL1b, rather than IL-1b transcription. This is presumably due to reduced expression of Ptgis and Ptgs1, and consequently lower levels of PGI2. PGI2, in turn, induces IL-10 expression that inhibits IL-1b processing, as well as TNF transcription. The manuscript is novel in its concept and findings, and upon answering the issues raised will constitute a significant contribution to the fields of immunology and cell signaling. However, significant flaws in experimental design and data interpretation should be rectified to increase its merit.

Major comments:

1. The results in EV3b are contrasting the main conclusion of the manuscript. They show that the production of pro-IL-1b in WT macrophages is inhibited by beraprost and IL-10, whereas in Fig. 4 beraprost and IL-10 did not affect the secretion of mature IL-1b. These results suggest that PGI2/IL-10 block the production of pro-IL-1b rather than just the processing of the protein. This conclusion is also supported by the reduced levels of Pro-IL-1b in unstimulated GATA6-deficient macrophages Fig. 2c and EV3. In addition, if the antibody used for flow cytometry binds the mature form of IL-1b then the terminology should be replaced to total IL-1b rather than pro. Also, adding the MFI values corresponding to the presented flow histograms will be helpful.
2. The data in EV3a shows 2 distinct cell populations in GATA6 deficient macrophages. One population expresses increased levels of pro-IL-1b both prior to and post LPS stimulation. The other does not express this cytokine even after LPS stimulation. The basis for these differences, which do not exist in WT macrophages is very interesting and should be explored. Are these differences coming from large vs. small peritoneal macrophages? Is there a leakage of expression in the GATA-6 KO? Also, unstimulated WT macrophages seem to express very high levels of pro-IL-1b that do not seem to match the values indicated by MFI in EV3b. Also, the histograms in b do not reflect the differences in MFI values shown for LPS and LPS+anti-IL-10 treatments in both genotypes.
3. The authors examined just one hypothesis for the findings they showed. That is PGI2 produced by GATA6 deficient macrophages is inducing IL-10 that inhibits IL-1b release. The potential contributions of PGE2, as an alternative inhibitory mediator, and TBXB2, as a stimulator of IL-1b expression were not explored. In addition, the potential regulation of IL-10 production through regulation of the PGI2 receptor, IP, by PGI2/PGE2/TBXB2 was not explored.
4. Along these lines, the authors used beraprost at 10 mM, which is about 100 times higher than its effective concentration on the IP receptor. At these concentrations EP4, DP and possibly other prostanoid receptors could be activated. The authors should use IP antagonists and additional IP agonists, like iloprost and treprostinil, at nanomolar concentrations to validate the involvement of PGI2-IP rather than PGE2-EP in the inhibitory action of GATA6-deficient macrophages.
5. Fig. 5 is very confusing in its presentation and the conclusion withdrawn. In b normalisation should be done to the unstimulated values rather than to the WT values. It is also not clear why 3 different dot plots are presented for each genotype in a. The MitoSOX staining should be shown in representative plots to explain the analysis that was done. It is perplexing that IL-10 and anti-IL-10 had the same effect on ROS production in GATA6-deficient macrophages. Since the results are negative in nature this reviewer think they should be omitted and indicated as data not shown.
6. According to the scheme presented in the manuscript, the second signal that activates the

inflammasome, in addition to LPS, should shut down the PGI₂/IL-10 mediated inhibition of IL-1 β and TNF production. This hypothesis should be examined experimentally to underscore the importance of the findings in this manuscript.

7. The authors indicated Etanercept did not affect IL-1 β secretion. However, the results in Fig. 1g indicate around 30% inhibition with statistical significance. This should be indicated in the results section. Also, whether Etanercept is interfering with the detection of TNF, rather than its production could easily be resolved by competition with recombinant TNF during the ELISA detection.

8. In Figure 2a, the mRNA data should be validated with flow cytometry for TLRs that are expressed in resident macrophages since differences in TLR4 expression could be due to differential surface expression between the genotypes.

9. In Figure 2c, densitometric analysis should be performed for pro-IL-1 β and the processed form to show statistical significance. It would also be beneficial to perform the experiments in Fig. 2C-D with release blockers to enhance the accumulation of the processed IL-1 β in GATA-6-deficient macrophages. In Fig. 2d quantification of the percentage of positive cells is not the informative readout. Rather, averages of MFI values should be presented as in EV3. It will be important to see if these values increase significantly after IL-1 β release inhibition.

10. In Fig. 2h-i, the inhibition of NLRP3 results in reduction in processed IL-1 β , but no increase in the pro-IL-1 β . Is this due to degradation of the precursor? Please explain in the text and/or examine experimentally. Also, the levels of caspase 1 protein and activity should be directly determined in both genotypes to conclude the inflammasome machinery is expressed at similar levels.

11. The results in Fig. 6b should be quantified by densitometry and graphed with statistical significance indicated. Also, the statistical significance lines should be aligned to prevent confusion. Moreover, the text in the results section indicate a reduction in NLRP3 levels by treatment with anti-IL-10 in both genotypes, while the graph shows only significance in WT macrophages. Please clarify. If indeed the increase in NLRP3 is responsible for the increased IL-1 β secretion following IL-10 neutralization, than using MCC950 with the antibodies should reduce IL-1 β levels. This should be shown experimentally.

Minor comments:

1. To verify the rigor of the data, results in presented all graphs should be repeated at least 3 times with data points indicated over the columns as in Fig. 1a. Different colors should be used for WT and GATA6-KO macrophages throughout the manuscript to enhance the clarity of presentation.

2. In the legend of figure 1 it should write IL-1 β (e) and TNF (f). Also, the receptor specificity of the TLR ligands should be indicated.

3. Please correct western to Western throughout the manuscript.

4. In all experiments that used anti-IL-10, the effect of the isotype control should be shown as well.

5. The summary of Fig. 6 indicates only the processing of IL-1 β is affected by the release of IL-10 following LPS stimulation in WT macrophages. However, the results presented show inhibition of IL-1 β and NLRP3 mRNA expression. Please correct accordingly.

6. Is the difference in IL-10 mRNA (Fig. 6) produced by GATA6-KO macrophages following LPS alone

We thank the referees for their positive and interesting comments and are pleased that they find our work novel and interesting. We believe their questions and advice have helped us improve the manuscript and we hope they agree.

For clarity, as we have made a number of changes to the data presentation during the revision process, which we recognise could cause confusion, we have summarised the main changes in an appendix at the end of this point-by-point reply.

Referee#1:

This is a very interesting study identifying a novel anti-inflammatory pathway in large Gata6 peritoneal macrophage. The mechanism involves PGI2 and IL10. I have a few questions to help give this *in vivo* relevance.

1) In the peritoneum there are large and small peritoneal cavity macrophage. Do the large regulate the small? If so how. Does the PGI2 and IL10 directly affect monocyte derived macrophage recruitment, cytokine production etc.

Author's reply: This is an interesting question and whilst we cannot provide an absolute answer, we can hypothesize that myeloid cells in the peritoneal cavity may interact with each other through this PGI2:IL-10 pathway. As noted by the reviewer, the cells within the peritoneal cavity of mice can be divided into two major populations: the larger tissue-resident peritoneal MΦ (Gata6⁺) and the smaller monocyte-derived MΦ. The deletion of Gata6 drastically decreases the numbers of resident peritoneal MΦ and we previously showed that Gata6-KO^{mye} mice have increased numbers of the MΦ/DC-like cells¹. These data suggest that Gata6 controls, directly or indirectly, the amount of small peritoneal MΦ under steady-state conditions.

We and others^{2,3} have also previously shown that the small MΦ population can be divided into further subpopulations, based on cell surface markers. We identified at least 4 subpopulations within the small MΦ category: the inflammatory MΦ (CD11b^{int} F4/80^{int} CD11c⁻ CD226⁻) and the "DC-like" or "MΦ/DC" cells that can be divided into CD11c^{low} CD226⁺, CD11c⁺ CD226⁺ and CD11c⁺CD226^{low}. Interestingly, we found naïve IL-10-deficient mice showed similar numbers of large peritoneal resident MΦ and small monocyte-derived cells, compared to IL-10^{+/+} mice (Fig. S7, Liao C.T. *et al*³), suggesting that, at steady state, IL-10 is not necessary for peritoneal MΦ differentiation and survival. However, when challenged with pro-inflammatory component, such as cell-free supernatant prepared from *S. epidermidis* (SES), the numbers of the four subtypes of small MΦ were differentially affected by the loss of IL-10, suggesting a subpopulation-dependant effect. It is important to note that *ex vivo* examination demonstrated the resident MΦ were the most prominent producers of IL-10 after SES challenge. We also observed that each myeloid cell population produces different level of cytokines (TNF, IL-6 and IL-12p40) after SES challenge (Fig. S5, Liao C.T. *et al*³), suggesting a possible differential response for each subpopulation in response to inflammation.

Given the potent effect of IL-10 on cytokine production⁴ and our present results, we expect that IL-10 treatment would reduce cytokine production *in vivo*, at least regarding IL-1β and TNF production. To date, the effect of PGI2 on peritoneal cells has not been carefully investigated but based on our current results, we might expect similar results to IL-10.

Cytokines measured in the peritoneal fluid may have been secreted from many cell types. More detailed discussion on this can be found in reply to point 3 below.

2) What happens in GATA6 deficient mice with LPS stimulation IP. This would help to elucidate the importance of this pathway in vivo. Are there alterations in neutrophil influx?

Author's reply: This is an interesting question, and we have now included the results of these experiments in the manuscript (new Fig. EV6). The analysis of such experiments in *Gata6-KO^{mye}* mice is challenging, given the basal differences in cell numbers compared to *Gata6^{fl/fl}* control mice. Naïve *Gata6-KO^{mye}* mice have a significantly decreased number of resident peritoneal MΦ, as well as an increased number of "MΦ/DC"-like cells and eosinophils. We injected the *Gata6-WT* and *Gata6-KO^{mye}* mice with intraperitoneal (i.p) injection of 1 ng LPS and observed no consistent difference in the absolute neutrophil counts 4 hours (h) after injection (new Fig. EV6a). This similarity in response is now added in the text. Finally, we could not see any effect of Anakinra (IL1R antagonist) treatment in a preliminary study of neutrophil recruitment 4 h after 1 ng LPS i.p injection (mice were pre-treated for 1 h with 50 mg/kg Anakinra before receiving the LPS, data shown below). This suggests that the early recruitment of neutrophils to the peritoneal cavity in this model is not dependant on IL-1β. This limited impact is perhaps not surprising, given that the defect is localised to a single cell population and the impact on IL-10 production by that cell type is also not complete. However, loss of IL-10 in acute inflammatory response such as this results in greater and sustained influx of neutrophils⁵.

3) Infectious agents tend to make these large peritoneal macrophage disappear. Although we still struggle to understand this macrophage disappearance reaction could the authors comment what happens after LPS delivery to the macrophage and cytokine production?

Author's reply: The question of resident MΦ disappearance during inflammation is very interesting. 1 ng of LPS is sufficient to induce inflammation, characterised by rapid neutrophil influx (see results above, in point 2), but it causes minimal resident MΦ loss and we only observed statistical differences between the *Gata6-WT* and *KO^{mye}* cells, those present basally (data shown below). We believe that the disappearance of the large resident MΦ is dependent on the type and amount of inflammatory agent, as exemplified by the

marked disappearance we have previously reported in a very low dose zymosan inflammatory response by comparison to the LPS response reported here⁶.

Regarding the production of cytokines *in vivo*, we did observe significant increases in pro-IL-1 β production by pM Φ of both genotypes after LPS injection and in the peritoneal lavage fluid, where it was increased in the Gata6-KO^{mye} 3 h after i.p. injection of 1 ng LPS compared to the Gata6-WT mice. However, given the basal differences between Gata6-KO^{mye} and -WT mice, we did not want to overinterpret these observations (Fig EV.6b and c). TNF was undetectable in the peritoneal fluid at that time point.

4) This sentence in the discussion and the reference are not correct. "Recently, Uderhardt and colleagues showed that pM Φ actively attempt to suppress inflammation and initiate local injury repair (26)." In this paper Germain and colleagues were not looking at the GATA6+ peritoneal macrophages but rather the tissue macrophages in the abdominal wall. Moreover, they simply show that the macrophages cloak and there was no repair function since they were only looking at the death of a single cell. The repair function of peritoneal macrophages was shown 3 years earlier by Wang and colleagues in Cell (2016).

Author's reply: Thank you for pointing our mistake. We apologise and have corrected it in the main text.

Referee #2:

The study by Ipseiz et al describes a new GATA6-induced pathway in resident peritoneal macrophages that limits IL-1b release following TLR activation. The authors provide evidence that GATA-6 deficient macrophages release excessive amount of IL-1b compared to WT cells due to enhanced processing of pro-IL1b, rather than IL-1b transcription. This is presumably due to reduced expression of Ptgs1 and Ptgs2, and consequently lower levels of PGI2. PGI2, in turn, induces IL-10 expression that inhibits IL-1b processing, as well as TNF transcription. The manuscript is novel in its concept and findings, and upon answering the issues raised will constitute a significant contribution to the fields of immunology and cell signaling. However, significant flaws in experimental design and data interpretation should be rectified to increase its merit.

Major comments:

1. The results in EV3b are contrasting the main conclusion of the manuscript. They show that the production of pro-IL-1b in WT macrophages is inhibited by beraprost and IL-10, whereas in Fig. 4 beraprost and IL-10 did not affect the secretion of mature IL-1b. These results suggest that PGI2/IL-10 block the production of pro-IL-1b rather than just the processing of the protein. This conclusion is also supported by the reduced levels of Pro-IL-1b in unstimulated GATA6-deficient macrophages Fig. 2c and EV3. In addition, if the antibody used for flow cytometry binds the mature form of IL-1b than the terminology should be replaced to total IL-1b rather than pro. Also, adding the MFI values corresponding to the presented flow histograms will be helpful.

Author's reply: We apologize if any of the data we presented were unclear. Firstly, all flow cytometry analysis of pro-IL-1 β have been performed with a specific antibody recognizing only the pro-IL-1 β unprocessed form (IL-1 beta (Pro-form) Monoclonal Antibody (NJTEN3), PE, eBioscience™, sold by Thermofisher, catalog number 12-7114-82), we have ensured that the antibody clone used is indicated in the manuscript.

Regarding the comment that “The results in EV3b are contrasting the main conclusion of the manuscript.” Note the data, which has been expanded upon and additional controls have been included, is now presented in new Fig6. We can say that IL-10 impacts pro-IL-1 β production (in the new Fig. 6d), albeit partially, and this is consistent with the partial reduction of *Il1b* mRNA in response to IL-10 (new Fig. 6a). No clear role for beraprost at this level of regulation is evident at either the pro-IL-1 β (Fig. 6d) or mRNA level (Fig. 6a) with this increased experimental size. The main conclusion of the manuscript is, however, centred on a marked regulation of the aberrant IL-1 β release by PGI2 and IL-10, as evidenced with the Gata6-knockout cells, where the enhanced IL-1 β release seen in these cells was abrogated by both PGI2 (Fig.4c) and IL-10 (Fig. 4f) .

The reviewer notes that “Fig. 4 beraprost and IL-10 did not affect the secretion of mature IL-1b”. This may again be in reference to only wild type cells which release little IL-1 β . It has been shown that healthy mouse wild type M Φ do not normally produce significant amounts IL-1 β when in presence of a primary signal only (here LPS)⁷. However, the knockout cells, show a substantial reduction in mature IL-1 β release in response to both IL-10 and beraprost (discussed above), which differs to their only partial impact on the pro-form and the mRNA of IL-1 β (discussed above), indicating that beraprost and IL-10 are indeed controlling IL-1 β production at an additional level.

Lastly, the reviewer notes “This conclusion is also supported by the reduced levels of Pro-IL-1b in unstimulated GATA6-deficient macrophages Fig. 2c and EV3”. In this circumstance (unstimulated culture conditions), very little IL-10 can be detected (Fig. 4a) and the PGI2 signature cannot be detected (Fig. 3e). We cannot assume that the same regulatory mechanisms are present in the absence of an inflammatory cue, indeed we described it as a ‘reactive pathway’. This is especially true when the regulatory elements cannot be readily detected, so we do not see that this observation contradicts the manuscripts conclusion, which is based upon a dysregulation of inflammasome-mediated IL-1 β in response to an inflammatory stimulation. This is discussed to some extent also in the reply to point 2 below.

“Also, adding the MFI values corresponding to the presented flow histograms will be helpful.”

The MFI values for both EV 3a and EV 3b were included in the graphs of EV 3b. These are now present in the new Fig. 6d. We are sorry this was not clear.

2. The data in EV3a shows 2 distinct cell populations in GATA6 deficient macrophages. One population expresses increased levels of pro-IL-1b both prior to and post LPS stimulation. The other does not express this cytokine even after LPS stimulation. The basis for these differences, which do not exist in WT macrophages is very interesting and should be explored. Are these differences coming from large vs. small peritoneal macrophages? Is there a leakage of expression in the GATA-6 KO? Also, unstimulated WT macrophages seem to express very high levels of pro-IL-1b that do not seem to match the values indicated by MFI in EV3b. Also, the histograms in b do not reflect the differences in MFI values shown for LPS and LPS+anti-IL-10 treatments in both genotypes.

We apologize for the confusion caused here. The plots were not the most representative of the data that was shown aggregated from several experiments in the then Fig. EV 3b, where if anything the Gata6-KO^{mye} cells show less IL-1 β content and varied positivity between different cell preparations.

This data is now better captured in Fig. 2e and Fig. 6d, the latter of which is a large repeat of this experiment with large numbers of animals processed individually. It should be noted that the study of pro-IL-1 β of ‘unstimulated’ M Φ *in vitro* is difficult as it is routine to see background activation at varying levels in every experiment. To emphasize that there is a level of background ‘culture-induced’ pro-IL-1 β , we have included in the new Fig. 6d (far right of the graph) flow-cytometric assessment of pro-IL-1 β in freshly isolated uncultured cells, and there is consistently no pro-IL-1 β in cells of either genotype. Fig. 2e shows data from multiple independent experiments highlighting the large variation in the percentage of ‘unstimulated’ M Φ that make pro-IL-1 β in culture. The original Fig. 2c (now Fig. 2d) shows a single Western Blot analysis of IL-1 β levels in cell lysates and this is now quantified in multiple experiments in Fig. EV4a, which shows no consistent difference between the genotypes. The new Fig. 6d consists of a large group of mice performed on the same occasion and shows a significantly reduced pro-IL-1 β production in Gata6-WT cells treated with IL-10 and LPS, compared to LPS alone. When blocking IL-10R, both WT and KO cells produced significantly more IL-1 β , compared to LPS only treated cells. However, these results do not contradict the conclusion of the manuscript. When stimulated with LPS, pro-IL-1 β appears relatively comparable between the two genotypes, and yet, the Gata6-deficient cells release substantially more IL-1 β in the absence of normal PGI₂ and IL-10 response, both of which can suppress the aberrant release of IL-1 β (discussed above).

3. The authors examined just one hypothesis for the findings they showed. That is PGI₂ produced by GATA6 deficient macrophages is inducing IL-10 that inhibits IL-1b release. The potential contributions of PGE₂, as an alternative inhibitory mediator, and TBXB₂, as a stimulator of IL-1b expression were not explored. In addition, the potential regulation of IL-10 production through regulation of the PGI₂ receptor, IP, by PGI₂/PGE₂/TBXB₂ was not explored.

We thank the referee for this interesting point. Firstly, with respect to the prostanoid receptors, this is very difficult to explore and we deliberately did not implicate IP in the manuscript, because the receptors and ligands are so promiscuous (please see more details in point 4) and the inhibitors lack the required specificity. However, we do not feel that this detracts from the novelty of the message of a prostanoid-IL-10 mediated regulation of IL-1 β release. With regard to PGE2 functioning in a similar manner to PGI2, whilst we did not explore this in the initial submission, because PGI2 was much more prevalent, we have now examined this more thoroughly and can confirm that PGE2 can suppress IL-1 β release and promote IL-10. This data is now included in Fig. 4. Attempts were made to block both thromboxane synthase and receptor with picotamide or antagonise the receptor with U46619 (both now shown in Fig. 4) in Gata6-KO cells to investigate the role of thromboxane signalling, but no differences were evident.

4. Along these lines, the authors used beraprost at 10 mM, which is about 100 times higher than its effective concentration on the IP receptor. At these concentrations EP4, DP and possibly other prostanoid receptors could be activated. The authors should use IP antagonists and additional IP agonists, like iloprost and treprostinil, at nanomolar concentrations to validate the involvement of PGI2-IP rather than PGE2-EP in the inhibitory action of GATA6-deficient macrophages.

Author's reply: We thank the reviewer for drawing our attention to this concern. For the experiments performed in this manuscript, we used a dose of 10 μ M of beraprost, not 10 mM indicated in the question. We assume the reviewer meant 10 μ M and not 10 mM and this is the consequence of a file error or similar. Please note that, in addition, we have now included data comparing the use of 1 μ M and 10 μ M beraprost (Fig. 4).

We initially based the concentration of beraprost we used on previously published work^{8,9}. The specificity of beraprost binding is indeed an interesting question. Beraprost has been shown to bind IP (PGI2 receptor) but also other receptors such as EP3 and EP4 (PGE2 receptors)^{10,11}, however with lower affinity (about 10-fold). Previous work have also shown that PGI2 itself has the capacity to bind TP (Thromboxane A2 receptor), EP1 and EP3¹², suggesting a dual role of PGI2. Both iloprost and treprostinil have been shown to have potent binding capacity to various receptors such as EP1, EP2, EP4 and DP1¹². Several reviews summarize the binding affinity of prostanoid agonists^{12,13}.

Based on the summary of Clapp and Gurung¹², beraprost seems to be one of the closest mimics of PGI2 in mouse. To evaluate the binding specificity of beraprost in our experiments, we performed a dose-dependent experiment (now shown below). We based our experiments on previously published doses of beraprost. It appears that lower doses of beraprost (down to 0.1 μ M) demonstrate significant inhibition of IL-1 β release after LPS stimulation. The progressive increase in IL-1 β production following beraprost decrease is correlated by the progressive decrease of IL-10 production, matching our hypothesis that beraprost induces IL-10, which in turn inhibits IL-1 β production. We have now also included experiments with iloprost (1 nM and 10 nM) and cicaprost (1 μ M and 10 μ M) in the manuscript (Fig. 4), which are consistent with the results obtained with beraprost.

It is important to note that due to the known limitations of the specificity of all the prostanoids and their analogues, with beraprost as well as PGI₂ able to bind not only IP but also EP₁, EP₃ and TP, we are not able to identify the receptor mediating the prostanoid effect we have observed nor did we make a claim about the receptor responsible for this regulatory role, but we do not feel this detracts from the proposed message of the manuscript.

5. Fig. 5 is very confusing in its presentation and the conclusion withdrawn. In b normalisation should be done to the unstimulated values rather than to the WT values. It is also not clear why 3 different dot plots are presented for each genotype in a. The MitoSOX staining should be shown in representative plots to explain the analysis that was done. It is perplexing that IL-10 and anti-IL-10 had the same effect on ROS production in GATA6-deficient MΦ. Since the results are negative in nature this reviewer think they should be omitted and indicated as data not shown.

We apologies if any of the data presented in Fig. 5 were confusing. We hope we have now improve the clarity of our figure, thanks to the referee's comments.

"In b normalisation should be done to the unstimulated values rather than to the WT values."

The data were normalised only to the WT unstimulated value, this preserves the relative differences between Gata6-WT and Gata6-KO^{mye} cells and any possible effect of the treatments.

"It is also not clear why 3 different dot plots are presented for each genotype in a."

We were simply showing the variability and have left just one set in the revised figure.

"The MitoSOX staining should be shown in representative plots to explain the analysis that was done. It is perplexing that IL-10 and anti-IL-10 had the same effect on ROS production in GATA6-deficient macrophages. Since the results are negative in nature this reviewer think they should be omitted and indicated as data not shown".

We have now removed the data as requested by the reviewer. To note though the IL-10 and anti-IL10 receptor treatments on LPS treated cells had the same effect on ROS production in that neither significantly affected it (when comparing to LPS treated alone).

6. According to the scheme presented in the manuscript, the second signal that activates the inflammasome, in addition to LPS, should shut down the PGI₂/IL-10 mediated inhibition of IL-1b and TNF production. This hypothesis should be examined experimentally to underscore the importance of the findings in this manuscript.

We do not necessarily agree that a second signal should shutdown the PGI₂/IL-10 mediated inhibition of IL-1 β and TNF. Compared to IL-1 β release, TNF regulation by PGI₂ and IL-10 appears to be a very different mechanism, primarily operating at the transcriptional level (Fig. 6e) and there is no reason to assume it is directly linked to the mechanism of inflammasome regulation. Activation of inflammasomes in both WT and Gata6-deficient M Φ elicits normal IL-1 β release in LPS primed, ATP or Nigericin stimulated cells (Fig. 2m) indicating that this regulatory mechanism is overcome by robust and rapid inflammasome stimulation.

We thank the referee for suggesting this experiment and we have now included it in Fig. 4j. It appears that IL-10 not only prevent the production of IL-1 β from LPS only stimulated cells (Fig. 4f), but also significantly reduces IL-1 β production when both LPS and a secondary stimulus (here ATP) are present (Fig. 4j).

7. The authors indicated Etanercept did not affect IL-1b secretion. However, the results in Fig. 1g indicate around 30% inhibition with statistical significance. This should be indicated in the results section. Also, whether Etanercept is interfering with the detection of TNF, rather than its production could easily be resolved by competition with recombinant TNF during the ELISA detection.

We appreciate the correction, our phrasing that etanercept “did not dramatically change” and did not “greatly affect” the IL-1 β release was clumsy and was intending to refer to the only partial impact of TNF inhibition and hence our rationale for focusing on the complete inhibition seen with beraprost and IL-10 as a stronger regulatory mechanism. We have revised this wording appropriately.

We speculated that etanercept may “block or prevent” detection of TNF production and this could occur at multiple levels, such as facilitated clearance of the TNF-etanercept stabilized complexes by the M Φ in culture, preventing detection. The cited manuscript provided a case of partial inhibition of human TNF detection via etanercept, as proof-of-principle of this concept, but equally the stabilisation of multimeric complexes with a different detection system could feasibly increase detection in a model system. We attempted to repeat the observations of this study with recombinant mouse TNF and saw at best a slight reduction in TNF detection (see below). Given the apparent absence of TNF detection, it seems that biological clearance of the TNF-etanercept complexes is more likely, but this (how etanercept:TNF complexes is cleared) is poorly studied and out of the scope of our current work. Consequently, given the uncertainty over this observation and what it may mean with respect to the limited impact of this treatment on IL-1 β release we have removed the TNF detection from the manuscript (Fig. 1).

8. In Figure 2a, the mRNA data should be validated with flow cytometry for TLRs that are expressed in resident macrophages since differences in TLR4 expression could be due to differential surface expression between the genotypes.

We thank the referee for the comment. We have now provided flow-cytometric validation for TLR4, CD14 and additionally TLR2 on freshly isolated resident MΦ. The data is now included in Fig. 2b and example plots are included in Fig. EV3. The Gata6-KO^{mye} resident MΦ exhibited higher surface expression of TLR2 and TLR4. However, as we do not see a difference in the pro-IL-1β (both on mRNA and protein level) produced in response to LPS, we hypothesise that the difference observed in TLR expression is not responsible for the higher secretion of the mature IL-1β by Gata6-KO^{mye} resident MΦ.

9. In Figure 2c, densitometric analysis should be performed for pro-IL-1b and the processed form to show statistical significance. It would also be beneficial to perform the experiments in Fig. 2C-D with release blockers to enhance the accumulation of the processed IL-1b in GATA-6-deficient macrophages. In Fig. 2d quantification of the percentage of positive cells is not the informative readout. Rather, averages of MFI values should be presented as in EV3. It will be important to see if these values increase significantly after IL-1b release inhibition.

Author's reply: we have now added the densitometric analysis of former Fig. 2c, now designated as Fig. 2d. The data can be found in Fig. EV4.

Blocking the release of the processed IL-1β would be something of great interest as it would help us answer some questions. However, the release mechanism of IL-1β is following a non-classical mechanism, independent of the conventional Golgi-pathway¹⁴. Although it is still a matter of debate, it appears that IL-1β is released either by exosomes, microvesicle shedding, cell lysis such as pyroptosis or through Gasdermin D pores in the cell membrane¹⁵⁻¹⁷. To our knowledge, there is not, to date, an efficient and specific way to block the release of processed IL-1β.

We apologize if the figure was unclear but the MFI values of pro-IL-1β⁺ macrophages were already added in fig. 2d (now fig. 2e). We hope that the graph is now more visible. The MFI for pro-IL-1β of total resident MΦ is now included in Fig. 6d.

10. In Fig. 2h-l, the inhibition of NLRP3 results in reduction in processed IL-1b, but no increase in the pro-IL-1b. Is this due to degradation of the precursor? Please explain in the text and/or examine experimentally. Also, the levels of caspase 1 protein and activity should be directly determined in both genotypes to conclude the inflammasome machinery is expressed at similar levels.

Author's reply: the IL-1 β analysis presented in the indicated figures (Western blot and ELISA) is on cell-free supernatants so we did not expect to, and did not, see pro-IL-1 β . Coll *et al.* showed in their original publication that MCC950 appears to have no effect on pro-IL-1 β ¹⁸. The additional data in our paper indicates similar levels of pro-IL-1 β in wild type and Gata6-deficient cells and it is unlikely that blocking cleavage of pro-IL-1 β would easily result in detection of increased pro-IL-1 β levels given the sensitivity of the assay.

11. The results in Fig. 6b should be quantified by densitometry and graphed with statistical significance indicated. Also, the statistical significance lines should be aligned to prevent confusion. Moreover, the text in the results section indicate a reduction in NLRP3 levels by treatment with anti-IL-10 in both genotypes, while the graph shows only significance in WT macrophages. Please clarify. If indeed the increase in NLRP3 is responsible for the increased IL-1b secretion following IL-10 neutralization, than using MCC950 with the antibodies should reduce IL-1b levels. This should be shown experimentally.

Author's reply: we have now added the densitometry analysis of the Western blot presented in Fig. 6b

We apologise that the significance lines were confusing and have done our best to improve them.

We confirm that treatment with anti-IL-10 receptor antibody in the presence of LPS increases Nlrp3 mRNA expression, significantly both in Gata6-WT and Gata6-KO^{mye} resident M Φ . The significance line had been accidentally omitted in the initial submission and this error has been now corrected. Please accept our apologies.

We thank the referee for the suggestion of the experiment. We have now performed it and included the results in Fig.4 h and i. When combined with anti-IL-10 antibody, MCC950 indeed decreased IL-1 β production by the Gata6-KO^{mye} cells, but not by the Gata6-WT cells. These data suggest that completely blocking IL-10 signalling during LPS stimulation of resident M Φ increases Nlrp3 more notably in WT cells (Fig. 6b), but the subsequent IL-1 β produced is likely Nlrp3 independent.

Minor comments:

1. To verify the rigor of the data, results in presented all graphs should be repeated at least 3 times with data points indicated over the columns as in Fig. 1a. Different colors should be used for WT and GATA6-KO macrophages throughout the manuscript to enhance the clarity of presentation.

Author's reply: all experiments have been now performed at least three times, as indicated in the figure legend and data presentation changed to reflect this.

We apology if the lack of colour impaired the clarity of the figures. We have modified all the graphs to include data points over the columns, and in addition coloured the columns using a colour-blind-friendly palette that matches the colours already used in some figures. We hope that the presentation of the figures is now clearer.

2. In the legend of figure 1 it should write IL-1 β (e) and TNF (f). Also, the receptor specificity of the TLR ligands should be indicated.

Author's reply: Thank you for noticing this. We have corrected these inaccuracies.

3. Please correct western to Western throughout the manuscript.

Author's reply: we have changed this, but it is our understanding that 'western' is not a proper noun, unlike Southern blot, which was named after Ed Southern. We do recognize, however, that this has become controversial and are happy to conform to whatever the editors wish to promote in their journal.

4. In all experiments that used anti-IL-10, the effect of the isotype control should be shown as well.

Author's reply: we assure the referee that the effect of the anti-IL-10 receptor antibody has been systematically analysed against the isotype. We have now included the isotype to the main figures or EV.

5. The summary of Fig. 6 indicates only the processing of IL-1b is affected by the release of IL-10 following LPS stimulation in WT macrophages. However, the results presented show inhibition of IL-1b and NLRP3 mRNA expression. Please correct accordingly.

Author's reply: we apologize for the lack of clarity in the summary, we have modified it now to clearly indicate 2 levels of regulation.

6. Is the difference in IL-10 mRNA (Fig. 6) produced by GATA6-KO macrophages following LPS alone

Yes, the difference in IL-10 produced by Gata6-KO M Φ in response to LPS alone is significant.

References

1. Rosas, M., *et al.* The transcription factor Gata6 links tissue macrophage phenotype and proliferative renewal. *Science* **344**, 645–648 (2014).

2. Bain, C.C., *et al.* Long-lived self-renewing bone marrow-derived macrophages displace embryo-derived cells to inhabit adult serous cavities. *Nat Commun* **7**, ncomms11852 (2016).
3. Liao, C.T., *et al.* IL-10 differentially controls the infiltration of inflammatory macrophages and antigen-presenting cells during inflammation. *Eur J Immunol* **46**, 2222–2232 (2016).
4. Moore, K.W., de Waal Malefyt, R., Coffman, R.L. & O'Garra, A. Interleukin-10 and the interleukin-10 receptor. *Annu Rev Immunol* **19**, 683–765 (2001).
5. Fielding, C.A., *et al.* Interleukin-6 signaling drives fibrosis in unresolved inflammation. *Immunity* **40**, 40–50 (2014).
6. Davies, L.C., *et al.* A quantifiable proliferative burst of tissue macrophages restores homeostatic macrophage populations after acute inflammation. *Eur J Immunol* **41**, 2155–2164 (2011).
7. Netea, M.G., *et al.* IL-1 β processing in host defense: beyond the inflammasomes. *PLoS Pathog* **6**, e1000661 (2010).
8. Chen, Y., *et al.* Prostacyclin analogue beraprost inhibits cardiac fibroblast proliferation depending on prostacyclin receptor activation through a TGF β -Smad signal pathway. *PLoS One* **9**, e98483 (2014).
9. Niwano, K., *et al.* Transcriptional stimulation of the eNOS gene by the stable prostacyclin analogue beraprost is mediated through cAMP-responsive element in vascular endothelial cells: close link between PGI₂ signal and NO pathways. *Circ Res* **93**, 523–530 (2003).
10. Fan, F., *et al.* Mechanism of Beraprost Effects on Pulmonary Hypertension: Contribution of Cross-Binding to PGE₂ Receptor 4 and Modulation of O₂ Sensitive Voltage-Gated K(+) Channels. *Front Pharmacol* **9**, 1518 (2018).
11. Gombert-Maitland, M. & Olschewski, H. Prostacyclin therapies for the treatment of pulmonary arterial hypertension. *Eur Respir J* **31**, 891–901 (2008).
12. Clapp, L.H. & Gurung, R. The mechanistic basis of prostacyclin and its stable analogues in pulmonary arterial hypertension: Role of membrane versus nuclear receptors. *Prostaglandins Other Lipid Mediat* **120**, 56–71 (2015).
13. Narumiya, S., Sugimoto, Y. & Ushikubi, F. Prostanoid receptors: structures, properties, and functions. *Physiol Rev* **79**, 1193–1226 (1999).
14. Stanley, A.C. & Lacy, P. Pathways for cytokine secretion. *Physiology (Bethesda)* **25**, 218–229 (2010).
15. Lopez-Gastejon, G. & Brough, D. Understanding the mechanism of IL-1 β secretion. *Cytokine Growth Factor Rev* **22**, 189–195 (2011).
16. Evavold, C.L., *et al.* The Pore-Forming Protein Gasdermin D Regulates Interleukin-1 Secretion from Living Macrophages. *Immunity* **48**, 35–44 e36 (2018).
17. Monteleone, M., Stow, J.L. & Schroder, K. Mechanisms of unconventional secretion of IL-1 family cytokines. *Cytokine* **74**, 213–218 (2015).
18. Coll, R.C., *et al.* A small-molecule inhibitor of the NLRP3 inflammasome for the treatment of inflammatory diseases. *Nat Med* **21**, 248–255 (2015).

Appendix: changes to data presentation structure:

Main manuscript data:

- Figure 1: is largely unchanged apart from the addition of a new panel to (h).
- Figure 2: includes a new part (b) (TLR expression, example data now in EV3), and new parts (j) and (k) (Caspase1 data).
- Figure 3: remains unchanged except for the addition of metabolites to (f) for better clarity.
- Figure 4: (a) remains unchanged
(b) and (c) are now comparing the effect of beraprost, cicaprost and iloprost at two concentrations on IL-10 and IL-1 β production in the presence of LPS.
(d) and (e) are now comparing the effect of beraprost, PGE2, U46619 and picotamide on IL-10 and IL-1 β production in the presence of LPS.
(f) correspond at former (c)
(g) is a repeated and expanded version of former (d)
(h) and (i) have been added to investigate the implication of Nlrp3 in our observations.
(j) has been added to investigate the effect of IL-10 treatment on IL-1 β release after primary (LPS) and secondary (ATP) stimulation
- Figure 5: as requested by the reviewer, this has been simplified and the exemplar data largely removed, as has the 'negative' mitosox data, which is now referenced in the text.
- Figure 6: (a) and former (c) (now e) has been repeated and expanded to include new controls
(b) now includes quantification of the Western data from the various experiments.
(c) and (d) are new and include include new caspase 1 activity data and pro-IL-1 β flow cytometric data (example raw data in new EV5)

Supplementary Data:

- Figure EV1: remains unchanged.
- Figure EV2: remains unchanged
- Figure EV3: example raw data that accompanies the new TLR data of Figure 2.
- Figure EV4: new Western blot quantification data of IL-1 β , Nlrp3 and Caspase1 expression
- Figure EV5: example flow cytometric data obtained from new repeat data with many mice (graphed in Figure 6).
- Figure EV6: new data showing the *in vivo* inflammatory response to LPS.

The reviewers should also note that in many cases we have overlaid scatter plots on bar graphs to better visualise the variability within the data.

Dear Prof. Taylor,

Thank you for submitting your revised manuscript to The EMBO Journal. Your study has now been seen by referee #2 and as you can see from the comments below the referee is happy with the introduced changes. S/he just have a few editorial comments.

I am therefore very pleased to let you know that we will accept your manuscript for publication here. Before doing so there are just a few additional points to take care off:

Regarding the transcriptome data of Gata6-WT and -KO mye pMΦ56 (page 8 data not shown). Since this is an important entry point into the prostacyclin part of this paper I think the transcriptome data should be provided. You can deposit it in a database and provide the accession number in the data availability section.

The reference format should be alphabetically with the first 10 authors listed before et al.

Would it be possible to combine 2 of the EV figures as we only allow for 5 EVs. The EV figures should be uploaded as individual figure files as well.

Please re-label declaration of Interests as COI

Our publisher has also done their pre-publication check on your manuscript and we have uploaded the manuscript file - called manuscript text -data edited. Please take a look at the word file and the comments regarding the figure legends and respond to the issues. Please also use this version when you resubmit the revised version with the marked changes. Just makes it easier for me to see the changes.

We include a synopsis of the paper (see <http://emboj.embopress.org/>). Please provide me with a general summary statement and 3-5 bullet points that capture the key findings of the paper.

We also need a summary figure for the synopsis. The size should be 550 wide by 400 high (pixels). You can also use something from the figures if that is easier.

That should be all - Congratulations on a great paper. Let me know if we need to discuss anything further

With best wishes

Karin

Karin Dumstrei, PhD
Senior Editor
The EMBO Journal

- a point-by-point response to the referees' comments, with a detailed description of the changes made (as a word file).

- a word file of the manuscript text.

- individual production quality figure files (one file per figure)

- a complete author checklist, which you can download from our author guidelines (<https://www.embopress.org/page/journal/14602075/authorguide>).

- Expanded View files (replacing Supplementary Information)

Further information is available in our Guide For Authors:

The revision must be submitted online within 90 days; please click on the link below to submit the revision online before 2nd Jul 2020.

Link Not Available

Referee #2:

The revised version of the manuscript is significantly improved. A few minor issues should be corrected.

Page 7, line 13- Replace choose with chose.

Page 7, last sentence, delete only.

Page 7, line 7, delete a.

Page 10, line 5, mye should be superscripted.

Fig. 6D, correct the color scheme.

We thank referee 2 for his useful comments.

We have corrected all the minor issues he indicated (pleased see in main manuscript).

Hi Phil,

Thanks for getting back to me regarding the "data not shown" statements in the MS text. Please proceed as discussed. You can upload the revised files via the link below

with best wishes

Karin

Karin Dumstrei, PhD
Senior Editor
The EMBO Journal

Link Not Available

We thank referee 2 for his useful comments.

We have corrected all the minor issues he indicated (pleased see in main manuscript).

Dear Phil,

Thanks for sending us the revised version. I have looked at everything and all looks good. I am therefore very pleased to accept the manuscript for publication here.

Congratulations on a nice study

with best wishes

Karin

Karin Dumstrei, PhD
Senior Editor
The EMBO Journal

Please note that it is EMBO Journal policy for the transcript of the editorial process (containing referee reports and your response letter) to be published as an online supplement to each paper. If you do NOT want this, you will need to inform the Editorial Office via email immediately. More information is available here: http://emboj.embopress.org/about#Transparent_Process

Your manuscript will be processed for publication in the journal by EMBO Press. Manuscripts in the PDF and electronic editions of The EMBO Journal will be copy edited, and you will be provided with page proofs prior to publication. Please note that supplementary information is not included in the proofs.

Should you be planning a Press Release on your article, please get in contact with embojournal@wiley.com as early as possible, in order to coordinate publication and release dates.

If you have any questions, please do not hesitate to call or email the Editorial Office. Thank you for your contribution to The EMBO Journal.

** Click here to be directed to your login page: <http://emboj.msubmit.net>

Corresponding Author Name: Philip R. Taylor

Manuscript Number: EMBOJ-2019-103454